# Tandem ketone reduction in pepstatin biosynthesis reveals an $F_{420}H_2$−dependent statine pathway

Jingjun Mo [1,2,7], Asfandyar Sikandar[1,7], Haowen Zhao [1,2,7], Ghader Bashiri [3], Liujie Huo [4], Martin Empting [1,5,6], Rolf Müller [1,2,5,6] & Chengzhang Fu [1,2] ✉

Pepstatins are potent inhibitors of aspartic proteases, featuring two statine residues crucial for target binding. However, the biosynthesis of pepstatins, especially their statine substructure, remains elusive. Here, we discover and characterize an unconventional gene cluster responsible for pepstatin bio-synthesis, comprising discrete nonribosomal peptide synthetase and polyke-tide synthase genes, highlighting its *trans*-acting and iterative nature. Central to this pathway is PepI, an $F_{420}H_2$-dependent oxidoreductase. The biochemical characterization of PepI reveals its role in the tandem reduction of β-keto pepstatin intermediates. PepI first catalyzes the formation of the central sta-tine, then produces the C-terminal statine moiety. The post-assembly-line formation of statine by PepI contrasts with the previously hypothesized bio-synthesis involving polyketide synthase ketoreductase domains. Structural studies, site-directed mutagenesis, and deuterium-labeled enzyme assays probe the mechanism of $F_{420}H_2$-dependent oxidoreductases and identify cri-tical residues. Our findings uncover a unique statine biosynthetic pathway employing the only known iterative $F_{420}H_2$-dependent oxidoreductase to date.

Pepstatins are highly potent inhibitors, effective in the pico- to nano-molar range, against various aspartic proteases (APs) such as pepsin and cathepsin D[1-5]. APs are endopeptidases discovered from all three domains of life and viruses. They are widely considered promising therapeutic targets owing to their vital roles in underlying physiolo-gical processes and the pathogenesis of various diseases. Notable examples include HIV-1 protease for HIV/AIDS, renin for hypertension, β- and γ-secretases for Alzheimer's disease, plasmepsin for malaria, and the secreted APs for fungal infections[6]. Pepstatins are linear *N*-acyl pentapeptides featuring two (3*S*,4*S*)−4-amino-3-hydroxy-6-methyl-heptanoic acid residues, which are nonproteogenic γ-amino acids known as statines (Sta; Fig. 1a). Crystal structures of different APs in

complex with pepstatins or structural analogs revealed that the central Sta residue forms hydrogen bonding with the two catalytic Asp residues[7], vital to AP inhibition (Fig. 1b). Sta mimics the tetrahedral transition state during catalysis forming a non-cleavable structural analog of the scissile bond present in AP substrates. Insights into the inhibitory mechanism of pepstatin on APs have been instrumental in the development of peptide isostere classes of AP inhibitors[8].

Since the discovery of pepstatins from actinomycetes[9,10], numer-ous studies have focused on their AP-inhibitory activity and mechan-ism. However, the pepstatin biosynthetic pathway has remained unclear. Pepstatins contain a Val-Val-Sta-Ala-Sta pentapeptide deco-rated with a variable *N*-terminal acyl chain[2,10,11]. The Sta residues in the

[1]Helmholtz Institute for Pharmaceutical Research Saarland (HIPS), Helmholtz Centre for Infection Research (HZI), Saarbrücken, Germany. [2]Helmholtz Inter-national Lab for Anti-Infectives, Helmholtz Center for Infection Research, Braunschweig, Germany. [3]Laboratory of Microbial Biochemistry and Biotechnology, School of Biological Sciences, University of Auckland, Private Bag, Auckland, New Zealand. [4]State Key Laboratory of Microbial Technology, Helmholtz International Lab for Anti-Infectives, Shandong University, Qingdao, China. [5]German Centre for Infection Research (DZIF), Braunschweig, Germany. [6]Department of Pharmacy, Saarland University, Saarbrücken, Germany. [7]These authors contributed equally: Jingjun Mo, Asfandyar Sikandar, Haowen Zhao. ✉e-mail: chengzhang.fu@helmholtz-hips.de

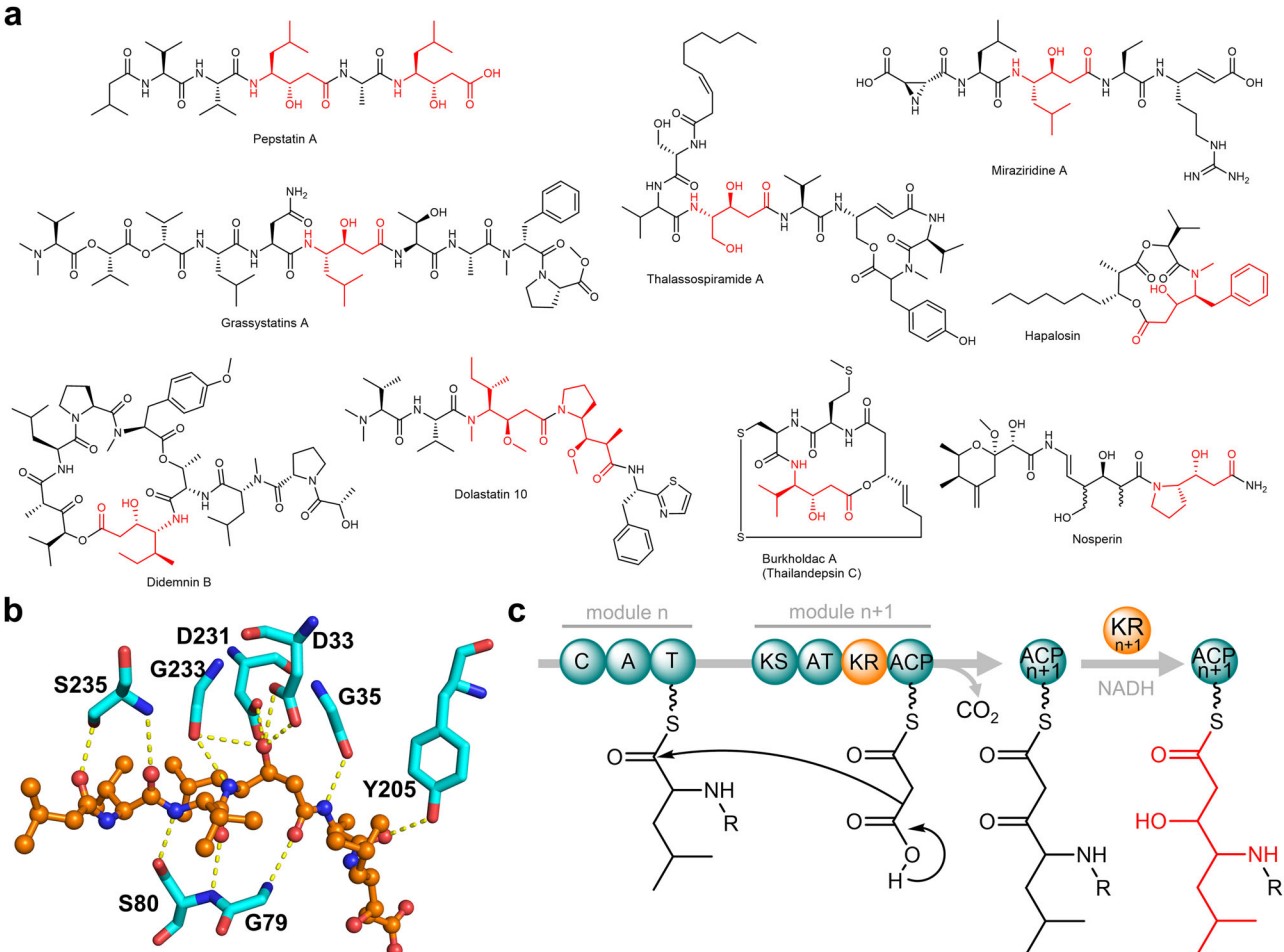

**Fig. 1 | Representative statine-containing natural products, pepstatin binding mode, and the proposed pathway for statine biosynthesis. a** Pepstatin A and representative known compounds with Sta/Sta-like residues (highlighted in red). **b** Interactions between pepstatin A and cathepsin D (PDB ID:1LYB). Only hydrogen bonds between pepstatin A (orange) and residues in the protein binding pocket (cyan) are shown. **c** The prior hypothesis on statine biosynthesis involving a modular PKS KR domain[26].

pepstatin-like ahpatinins may be replaced by 4-amino-3-hydroxy-5-phenylpentanoic acid but retain the 3-OH-4-$NH_2$ feature[12,13]. A feeding study using [14]C-labeled compounds suggested the Sta residues in pepstatin are derived from L-leucine and malonate[14]. The Sta or Sta-like residues with the 3-OH-4-$NH_2$ framework are also found in other natural products (representative examples in Fig. 1a)[15–25]. The biosynthesis of these substructures has been hypothesized to involve a collaborative process between nonribosomal peptide synthetase (NRPS) and polyketide synthase (PKS) modules[26]. In this proposed pathway, an NRPS module activates an α-amino acid such as leucine and loads it to the cognate peptidyl carrier protein (PCP) domain. The acyltransferase (AT) domain then attaches the malonyl-CoA extending unit to the acyl carrier protein (ACP) domain. A Claisen condensation reaction leads to the extension of the peptidyl thioester, forming a β-keto ester. In the didemnin biosynthetic gene cluster (BGC)[27], isoleucine is activated, and the ketoreductase (KR) domain within the PKS module reduces the β-keto group to form the 3-OH-4-$NH_2$ framework (Fig. 1c)[26]. A similar NRPS-PKS pair with a KR domain in the PKS module has also been observed in the BGCs of burkholdac[23], hapalosin[28], thalassospiramide[29], and nosperin[21]. In contrast, the β-ketone functionality is retained in andrimid biosynthesis. Notably, the corresponding NRPS-PKS pair in the andrimid biosynthetic pathway lacks a KR domain, implying that the KR domain in PKS modules is responsible for 3-OH generation in Sta/Sta-like residue biosynthesis[30]. However,

this pathway has not been experimentally verified and has only been rationalized based on the analysis of BGCs of natural products with the 3-OH-4-$NH_2$ frameworks[26].

In this work, we uncovered a unique post-assembly-line pathway for Sta biosynthesis that relies on a discrete oxidoreductase utilizing $F_{420}H_2$, a deazaflavin-based redox cofactor widely found in bacteria and archaea[31–33]. This pathway diverges from the conventional hypothesis, which attributes this function to a modular KR domain utilizing NADPH (Fig. 1c). Through gene knockout experiments and activation of the candidate pepstatin BGC, we characterized a non-canonical NRPS-PKS pathway responsible for pepstatin biosynthesis. Our findings suggest the involvement of in-trans and iterative NRPS and PKS mechanisms, indicating deviations from the traditional colinearity principle[34,35]. Deletion of *pepI*, encoding an $F_{420}H_2$-dependent oxidoreductase, resulted in an accumulation of the β-keto intermediates of pepstatins and related spontaneously decarboxylated products. Biochemical characterization of PepI in vitro unambiguously demonstrated the sequential reduction of β-ketones, starting with the central residue and then followed by the C-terminal moiety, ultimately leading to the formation of pepstatins. We determined the specific structural changes in the substrates and products of PepI through comprehensive structure elucidation and enzymatic characterization. The crystal structures of PepI and PepI in complex with the cofactor $F_{420}$ allowed us to identify key residues His62, Tyr122, and Gln229. The

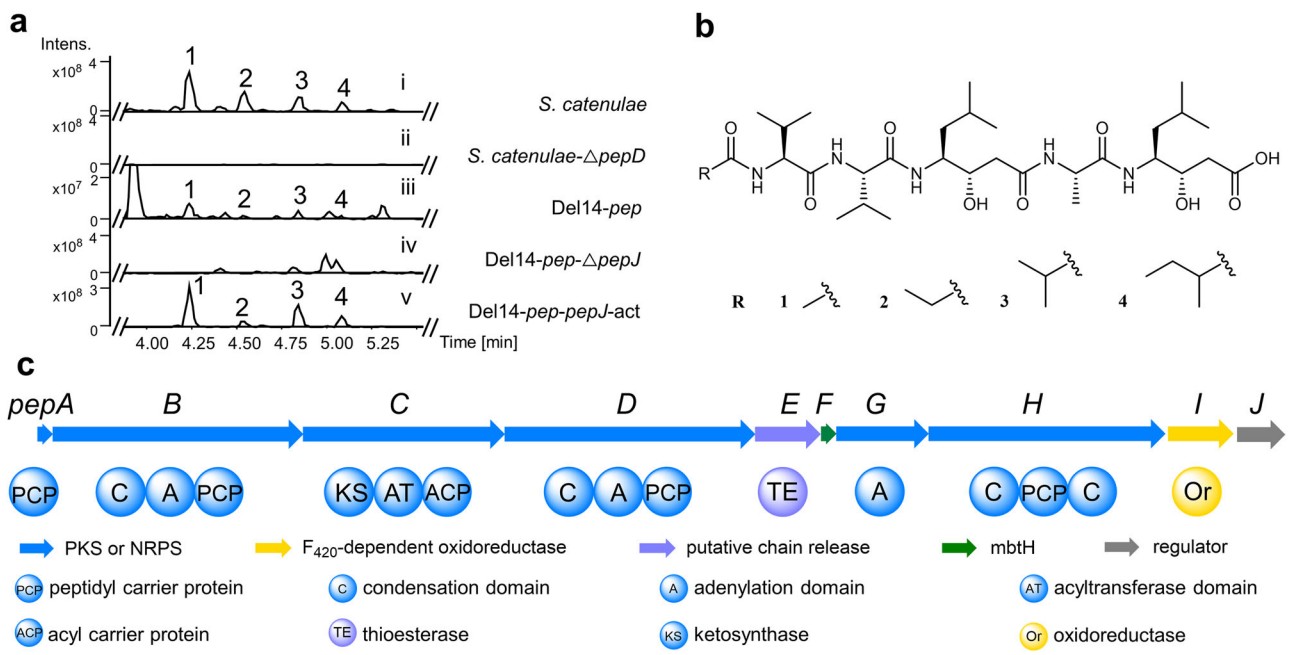

**Fig. 2 | Identification of pepstatins and the *pep* biosynthetic gene cluster.**
**a** UPLC-HRMS analysis (base peak chromatogram (BPC)) of pepstatin congeners (**1**-**4**) produced by *Streptomyces catenulae* DSM40258 (i); Knockingout *pepD* abolished pepstatin **1**-**4** production (ii); Pepstatin congeners (**1**-**4**) produced by heterologous expression of BGC *pep* in Del14 (Del14-*pep*) (iii); The production of **1**-**4** significantly decreased by *pepJ* deletion in Del14-*pep*-Δ*pepJ* (iv); The production of **1**-**4** increased in Del14-*pep*-*pepJ*-act by promoter exchange of *pepJ* (v). **b** The chemical structures of pepstatins isolated from *Streptomyces catenulae* DSM40258. **c** The schematic representation of the pepstatin BGC from *S. catenulae* DSM40258.

roles of these residues are further rationalized by PepI-substrate modeling and site-directed mutagenesis studies. Moreover, biochemical assays utilizing a deuterium-labeled cofactor provide valuable insights into the catalytic mechanism of $F_{420}$-dependent oxidoreductases. Notably, PepI-mediated Sta biosynthesis represents the only known iterative $F_{420}H_2$-dependent oxidoreductase, distinct from previously hypothesized PKS KR-dependent biosynthetic pathways for 3-OH-4-$NH_2$ frameworks.

## Results

### Identification of the pepstatin BGC defying the colinearity rule

We cultivated the known pepstatin producer *Streptomyces catenulae* DSM40258[36] and analyzed its fermentation broth using ultra-performance liquid chromatography-high-resolution mass spectrometry (UPLC-HRMS). Structural elucidation of the four purified pepstatin congeners (**1**-**4**) through nuclear magnetic resonance (NMR) spectroscopy revealed variations at the *N*-terminal acyl group, confirming congener **4** is a new derivative (Fig. 2a, b; Supplementary Figs. 1 and 26–42; and Supplementary Table 8)[10,11].

With pepstatin production verified, we performed complete genome sequencing of *S. catenulae* DSM40258 to decipher the BGC of pepstatin. Surprisingly, the absence of a straightforward candidate for an NRPS-PKS hybrid BGC suggested a noncanonical biosynthetic pathway. According to the colinearity principle, we expected to find five NRPS modules and two PKS modules in one BGC (Supplementary Fig. 2). However, none of the NRPS-related BGCs encoded more than four adenylation (A) domains, indicating an unconventional biosynthesis mechanism. Furthermore, we found no consecutive NRPS and PKS modules that matched the previously reported biosynthesis of statine/statine-like residues in the DSM40258 genome. Despite these unexpected challenges, we identified one NRPS-PKS hybrid BGC (designated as "*pep*") with only three A domains that caught our attention. This BGC included one putative leucine-activating NRPS gene (*pepB*) adjacent to a PKS gene (*pepC*) with a truncated AT domain

(Fig. 2c, Supplementary Table 5). To confirm the involvement of the *pep* BGC in pepstatin biosynthesis, we performed gene deletion of *pepD*, which encodes the potential alanine assembly module. Deleting the NRPS gene *pepD* completely abolished pepstatin production in *S. catenulae* DSM40258 (Fig. 2a; Supplementary Figs. 3 and 7a), demonstrating that *pepD* is essential for pepstatin biosynthesis.

To validate the biosynthetic pathway, the 18.3 kb *pep* BGC, comprising ten genes (*pepA-J*), was cloned and subsequently heterologously expressed in *Streptomyces albus* Del14[37]. The successful heterologous reconstitution of pepstatin production confirmed the BGC and its boundaries (Fig. 2a and Supplementary Fig. 4). Deletion of the LuxR-like transcriptional regulator gene *pepJ* significantly reduced pepstatin production, suggesting that *pepJ* functions as a positive regulator. Conversely, overexpression of *pepJ* under the *kasOp* promoter led to a more than 25-fold increase in pepstatin yield (Fig. 2a and Supplementary Figs. 5–7), enabling the identification of biosynthetic intermediates in subsequent studies.

### A highly dissociated and nonlinear NRPS-PKS pathway for pepstatin biosynthesis

The streamlined *pep* BGC presents an interesting puzzle, as it contains fewer NRPS-PKS modules than the number of building blocks required for pepstatin biosynthesis, suggesting the involvement of noncanonical mechanisms. The *pep* BGC possesses only three A domains to assemble the pentapeptide backbone of pepstatins (Fig. 2c), which deviates from the colinearity principle. However, based on the chemical structure, pepstatin assembly hypothetically requires only three types of amino acids: a single L-alanine residue, and two occurrences each of L-valine and L-leucine (the precursor of statine), suggesting iterative use of A domains in its biosynthesis.

The A domain from the single-module NRPS gene *pepD* is predicted to activate alanine, while PepG, the stand-alone A domain protein, is predicted to activate threonine (Supplementary Table 5). PepH features an unusual C-PCP-C domain arrangement without any

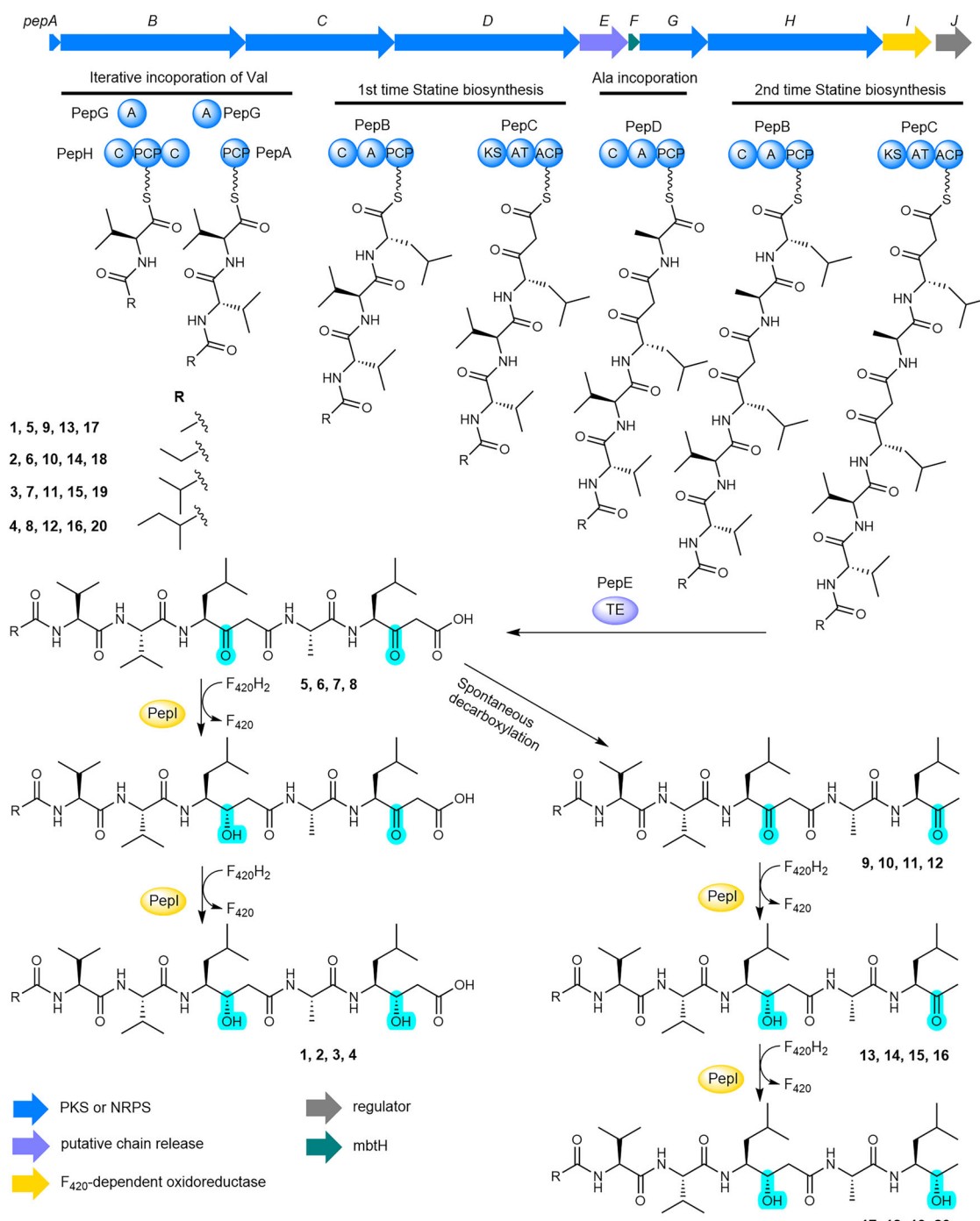

**Fig. 3 | Proposed pepstatin biosynthesis pathway.** Fatty acids are likely activated to fatty acyl-CoA by acyl-CoA synthase and transferred to the carrier protein PepA, whereas PepG, PepH, PepB, PepC and PepD build up the peptide chain. The pentapeptide chain is subsequently released by PepE. Oxidation status changes are highlighted in turquoise to exhibit the two ketoreduction reactions catalyzed by PepI.

cognate A domains. PepG is likely promiscuous in substrate activation, capable of activating threonine and similar amino acids, including valine. However, the C domains in PepH might serve as gatekeepers, processing only proteins loaded with valine[38]. We propose that pepstatin biosynthesis initiates with the activation and in-trans loading of valine by PepG onto the PCP domain of PepH. The $N$-terminal C domain of PepH then acylates the Val-$S$-PepH species using various short-chain fatty acyl CoAs. A plausible subsequent step involves PepG reactivating valine and loading it onto the stand-alone PCP protein PepA. The $C$-terminal C domain of PepH may then facilitate chain extension by

attacking the Val-$S$-PepA upon the thioester of the $N$-acyl-Val-$S$-PepH intermediate (Fig. 3).

The A domain of the single-module NRPS gene *pepB* is predicted to activate leucine, while *pepC*, the sole PKS gene, encodes a KS-AT-ACP module. These findings suggest that PepB and PepC may collaborate to assemble the Sta residue backbone in pepstatins, similar to previously proposed Sta biosynthesis (Fig. 1c). However, the absence of a KR domain in PepB implies a remarkable new feature in Sta biosynthesis. We propose that PepB extends the nascent $N$-acyl-Val-Val chain with leucine, and PepC subsequently installs a malonyl-CoA to

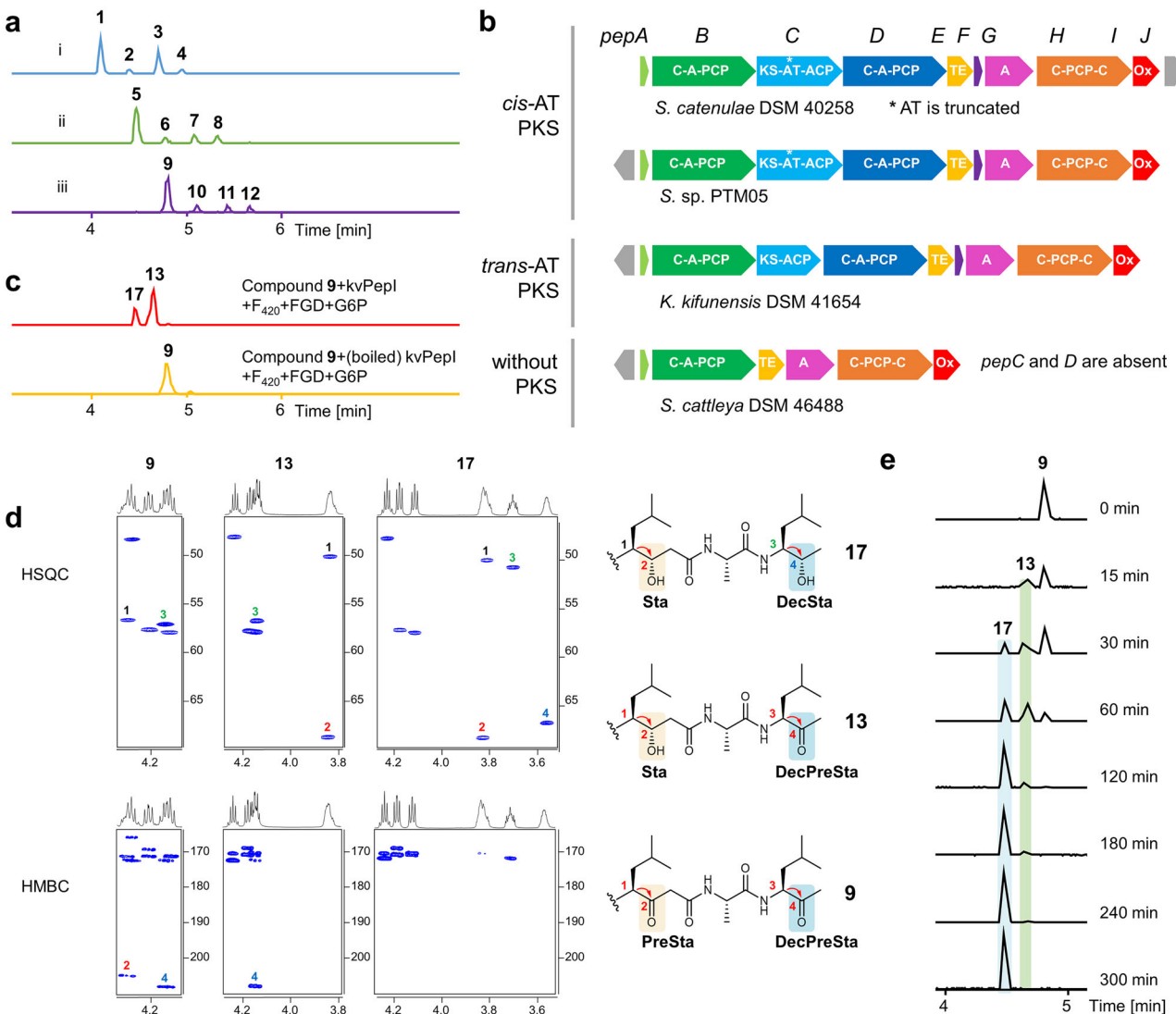

**Fig. 4 | PepI catalyzes tandem ketone reductions. a** UPLC-HRMS analysis of pepstatins and unreduced β-keto intermediates in the *pepI* deletion mutant. (i) EICs ([M + H]⁺, blue) of **1** (644.42), **2** (658.44), **3** (672.45), **4** (686.47) from the improved heterologous expression of *pep* (*pepJ*_act); (ii) EICs ([M + H]⁺, green) of **5** (640.39), **6** (654.40), **7** (668.42), **8** (682.44) and (iii) EICs ([M + H]⁺, purple) of **9** (596.40), **10** (610.42), **11** (624.44), **12** (638.45) from the *pepI* deletion mutant based on Del14-*pep*-*pepJ*-act. **b** Representative *pep*-like actinobacterial pathways classified into three types mainly distinguished by containing *cis*-AT PKS, *trans*-AT PKS, or without PKS.

**c** In vitro characterization of kvPepI. EICs ([M + H]⁺, red) of **13** (598.42) and **17** (600.43) produced by kvPepI reaction with **9** and EIC ([M + H]⁺, orange) of **9** from the control reaction using boiled kvPepI and **9** were shown. **d** HSQC and HMBC slices of **9**, **13**, and **17**, showing the structural changes at the Sta residue, dec-arboxylated Sta residue (DecSta), the precursor of Sta (PreSta), and the precursor of DecSta (DecPreSta). **e** Time-course of kvPepI using **9** as the substrate, supple-mented with F₄₂₀, FGD, and G6P. BPCs of the reactions were shown.

form the β-ketoacyl species. PepD then appears to activate and incorporate alanine into the growing NRP-PK chain. Following this, PepB and PepC may collaborate again to extend the lipopeptide backbone with a second β-ketoacyl building block. This second extension involving PepB and PepC following PepD, an unrelated NRPS module, is highly unusual and partially similar to the pass-back mechanism observed in thalassospiramide biosynthesis[39]. Notably, in pepstatin biosynthesis, this extension likely involves inter-protein interactions (Fig. 3), representing an in-trans mechanism distinct from intramodule interactions observed in other systems[39]. Overall, while the proposed pathway outlines a plausible biosynthetic logic, addi-tional experimental data are needed to validate these hypotheses and clarify the unique enzymatic features of the pepstatin biosynthetic machinery.

The mechanism underlying the double reduction of the β-keto group during pepstatin biosynthesis remains unresolved due to the absence of a KR domain. The timing and nature of these unidentified reductions are particularly intriguing, given the presence of two Sta residues. We assumed it is likely that pepstatin biosynthesis, unlike other Sta-containing natural products, has evolved a unique β-keto reduction mechanism that may involve enzymes working either in tandem or sequentially. A prime candidate for this reduction process is PepI, a putative F₄₂₀-dependent oxidoreductase, which shares low sequence identity (26.1%) with the TIM-barrel type F₄₂₀-dependent methylenetetrahydromethanopterin reductase (Mer)[40].

## PepI is an F₄₂₀H₂-dependent ketone reductase
Deletion of the *pepI* gene in the heterologous expression system of the *pep* BGC abolished the production of **1-4**, while a series of new peaks emerged, corresponding to unreduced β-keto intermediates **5-8** as identified by MS/MS analysis (Figs. 3 and 4a and Supplementary Figs. 8 and 9). However, the β-ketoacid compounds **5-8** decomposed

rapidly through spontaneous decarboxylation into **9-12** (Fig. 4a), which lack the terminal carboxyl group, as confirmed by NMR analysis (Fig. 4d and Supplementary Figs. 10 and 43–62, Supplementary Table 9). Unlike the KR domain typically responsible for reduction during chain extension, the accumulation of the free β-keto intermediates (**5-12**) suggested that PepI functions as a tailoring enzyme, likely reducing the ketone groups at the β-position of the two β-diketone moieties after the intermediates are released from PepC (Fig. 3).

To investigate this distinct mechanism of statine formation, we aimed to characterize PepI in vitro. Recombinant PepI was successfully purified from *E. coli* BL21 (DE3). However, the recombinant protein exhibited unsatisfactory solubility (Supplementary Fig. 11b), precluding its use in downstream structural biology studies. To address this challenge, we searched public databases for homologs of PepI and identified several *pep*-like BGCs harboring *pepI* gene homologs. These BGCs are classified into three distinct subtypes, primarily based on variations in their PKS genes (Fig. 4b, Supplementary Fig. 11a, and Supplementary Table 6). The *pep* BGC represents the first subtype, characterized by a *cis*-AT PKS gene, while the second subtype features a *trans*-AT PKS gene that lacks the AT domain. Notably, all PKS modules in these BGCs are devoid of KR domains. We therefore hypothesize that PepI analogs serve as substitutes for KR domains to generate the 3-OH-4-NH₂ moiety. The third subtype, however, lacks homologs of PepC and PepD, raising questions about its ability to produce Sta/Sta-like residues containing products (Fig. 4b). Among these, the homolog kvPepI from *Kitasatospora viridis* DSM44826, with 53.3% sequence identity to PepI, was more amenable to study (Supplementary Fig. 11b). Searching *pepI* homologous genes in genomes of producers of didemnin[27], burkholdac[23], hapalosin[28], thalassospiramide[29], and nosperin[21] did not lead to any hit, further corroborating the unique Sta formation in pepstatin pathway.

We reasoned that **5-8** are likely the authentic intermediates for **1-4**, but only the decarboxylated products **9-12** were stable enough to be isolated in sufficient quantities for in vitro assays. We primarily used the most abundant congener, **9**, as the substrate in all PepI and kvPepI assays. To provide and regenerate the reduced cofactor $F_{420}H_2$, we supplemented the reaction mixture with $F_{420}$, the $F_{420}$-dependent glucose-6-phosphate dehydrogenase (FGD)[41], and its substrate glucose-6-phosphate (G6P). After overnight incubation with PepI or kvPepI, UPLC-HRMS analysis revealed the complete consumption of **9** and the appearance of a new single-charged ion with a 4 Da increase, indicating the reduction of two ketones in **9** (Fig. 4c, Supplementary Figs. 10 and 12–14). Subsequently, we scaled up the kvPepI enzymatic assay using **9**, which allowed us to purify new product **17** for NMR analysis, confirming the presence of the middle β-hydroxyl group and C-terminal secondary alcohol (Fig. 4d, Supplementary Figs. 68–72 and Supplementary Table 10).

We observed the same 4 Da mass shift using **10, 11**, or **12** as substrates for PepI or kvPepI, indicating consistent dual ketone reduction supported by MS/MS analysis (Supplementary Figs. 10 and 12–14). Since the authentic substrates **5-8** were too unstable for isolation due to the rapid spontaneous β-keto decarboxylation, we used the fermentation broth of the mutant Del14-Δ*pepI* for in vitro assays. As anticipated, the incubation with PepI resulted in the production of **1-4**, as detected by UPLC-HRMS (Supplementary Fig. 15). These results confirmed that PepI is the enzyme responsible for the dual ketone reduction in pepstatin biosynthesis. Removing $F_{420}$, FGD, or G6P abolished PepI or kvPepI activity, underscoring the essential role of $F_{420}H_2$ in these transformations (Supplementary Figs. 13 and 14).

### PepI catalyzes the tandem reduction of two ketones
With the role of PepI in reducing ketones during pepstatin biosynthesis confirmed, we focused on the timing of the two reduction steps. During our investigation, in addition to product **17**, we identified a previously uncharacterized intermediate, **13**, which was only 2 Da heavier than **9** (Fig. 4c). This led us to hypothesize that **13** could be an intermediate of **9** with only one of the ketones reduced. MS/MS and NMR analysis unambiguously determined the structure of purified **13**. The HSQC and HMBC correlations of **13** clearly indicated the presence of a β-hydroxyl group at the middle residue (Sta3), with the methyl ketone at the terminal position remaining unchanged (Fig. 4d, Supplementary Figs. 10, 16, 58–62 and Supplementary Table 10). These findings are also corroborated by MS/MS analysis of the corresponding intermediates with an additional 2 Da when using **10, 11**, or **12** as substrates (Supplementary Figs. 10 and 16).

To further examine the order of the two ketone reductions, we conducted time-course experiments using kvPepI. The time course-assay of kvPepI showed that **13** appeared first and reached the maximum concentration within three hours. Compound **17** emerged later than **13** and became apparent after one hour of incubation. The amount of **17** kept increasing for 5 h until **9** and **13** were nearly consumed completely (Fig. 4e). These results indicate that the reduction of the middle β-ketoamide moiety occurs before the reduction of the terminal ketone, establishing the sequential nature of PepI's catalytic activity.

### Structure of kvPepI
To better understand the mechanism underlying the reduction reaction, we determined the crystal structures of kvPepI and its complex with the cofactor $F_{420}$, both to a resolution of 1.65 Å (Supplemtary Table 7). We found that kvPepI is structurally similar to other members of Class I $F_{420}$-dependent enzymes (Fig. 5a, Supplementary Fig. 19a, b), including $F_{420}$-dependent alcohol dehydrogenase (Adf, PDB ID: 1RHC; Supplementary Fig. 19c, Cα RMSD over entire length of the protein: 2.9 Å)[42]. Upon $F_{420}$ binding, the kvPepI structure remains virtually unchanged, except for the stabilization of the residues 187–190 (Fig. 5a and Supplementary Fig. 20) that are part of a larger flexible loop (residues 181–201). This loop forms a lid over the lactyloligoglutamyl tail of $F_{420}$, likely anchoring the $F_{420}$ for catalysis. A number of hydrogen bonds and hydrophobic interactions further stabilize the cofactor (Supplementary Fig. 21a). Interestingly, the C5 of $F_{420}$ was found to be positioned within 3.8 Å of a water molecule, held in place by His62 and Tyr122 (HOH570; Supplementary Fig. 21b). Overlay of Adf-acetone adduct structure with kvPepI-$F_{420}$ revealed that the keto group of the acetone adduct occupies a position similar to that of the ordered water (HOH570, Fig. 5b, Supplementary Fig. 21a), suggesting that the highly conserved His62 acts as a general acid to protonate the keto group of **9**. The His imidazole moiety likely acts as a proton relay through substrate protonation via the His62-Nε2 position and interaction with the carboxyl group of the Glu126 sidechain via the His62-Nδ1 (Supplementary Fig. 21b). Accordingly, the H62A mutation abolished the PepI activity (Fig. 5d, e). To rule out the possibility of H62A mutation causing a loss of activity due to allosteric effects, we also solved the structures of kvPepI[H62A] and kvPepI[H62A] in complex with $F_{420}$, which are virtually identical to kvPepI apo and kvPepI-$F_{420}$ complex structures, respectively (Supplementary Fig. 21c and Supplementary Table 7). Taken together, these findings strongly support the role of His62 as the proton-donating residue.

### Substrate mode of binding and mutational analysis of kvPepI
Despite repeated attempts, we could not determine the structure of either kvPepI or kvPepI-$F_{420}$ in complex with **9**. Therefore, we decided to model **9** in the kvPepI-$F_{420}$ crystal structure using the gathered experimental data as well as the above-mentioned ordered water molecule to guide the position of the β-keto group of the oxidized statin precursor residue PreSta3 residue in **9** (Fig. 5b, c). For **9** two different binding modes were considered: linear and U-shaped conformation (Supplementary Fig. 22). However, only in the U-shaped conformation are two relatively well-conserved residues, Gln229 and

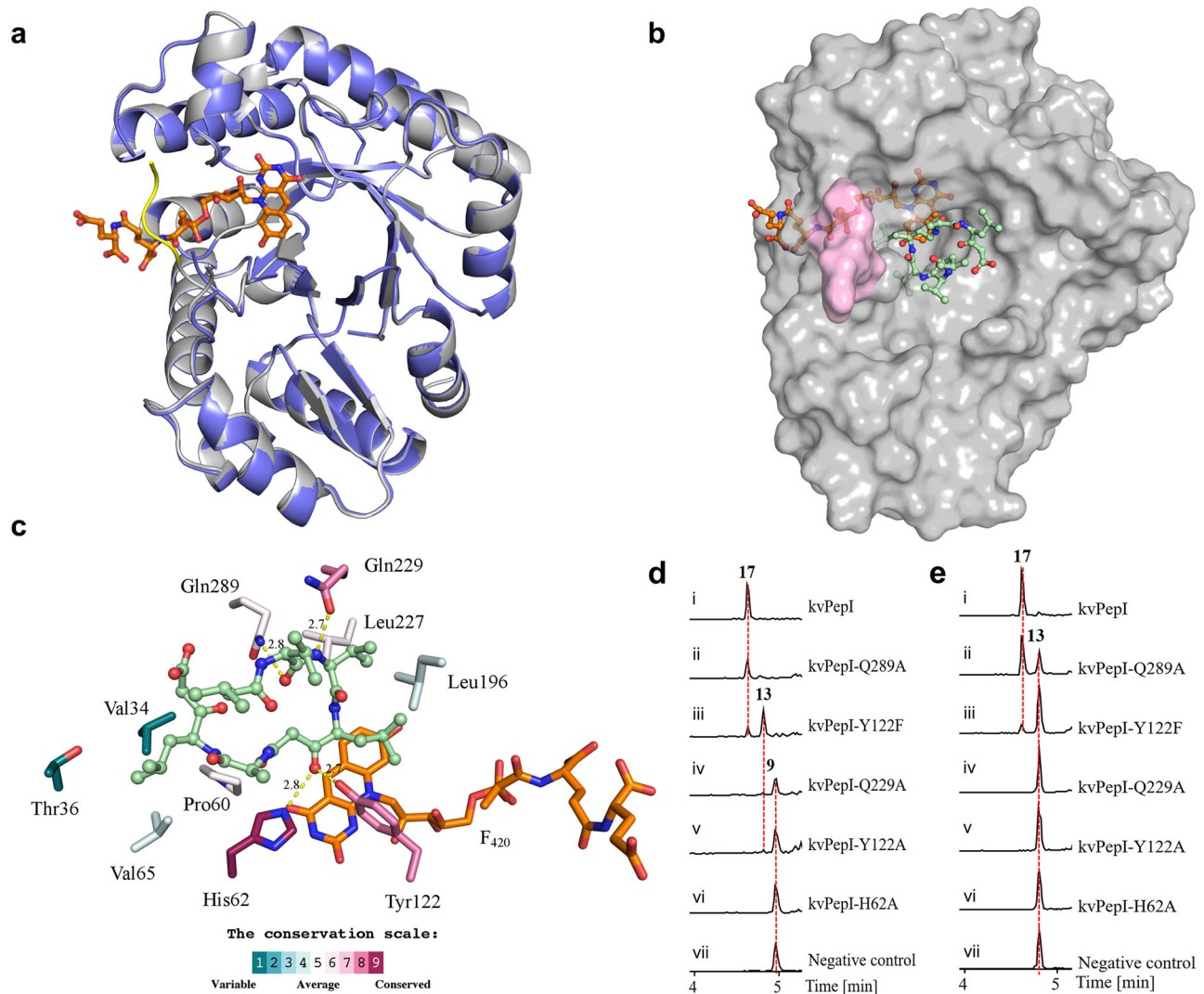

**Fig. 5 | Structural analysis, molecular modeling, and biochemical assays of kvPepI and mutants. a** Superposition of the kvPepI structure with (gray) and without (slate) $F_{420}$ bound. Changes in the overall structure of the protein are minimal (Cα RMSD of 0.15 Å over all non-hydrogen atoms), except for the stabilization of a loop (residues 187 – 190; yellow line) that serves as a lid over the bound cofactor. **b** Hypothetical binding pose of U-shaped **9** in kvPepI – $F_{420}$ complex structure. The missing loop (pink) was modeled using AlphaFold[68,69]. The orientation and conformation of **9** were modeled based on experimental data regarding the position of hydride transfer, as well as loss-of-activity-conferring amino acid mutations (details see experimental section). $F_{420}$ (orange) and **9** (pale green) are shown as sticks (atom color: carbon black, nitrogen blue, oxygen red). **c** Schematic 3D representation of kvPepI – $F_{420}$ interactions with modeled pose of **9**. Hydrogen bonds are depicted with dotted yellow lines with distances given in Å, while all other residues form hydrophobic interactions with the substrate. The residues are colored according to the conservation score calculated using the ConSurf server[74,75]. Reactions using **9** (**d**) or **13** (**e**) as the substrate with kvPepI (i), Q289A (ii), Y122F (iii), Q229A (iv), Y122A (v), H62A (vi), and boiled kvPepI as negative control (vii). BPCs of reactions were shown.

Gln289, located close to the terminal end of the U-shaped **9** in the binding pocket (Fig. 5c and Supplementary Fig. 23c), forming hydrogen interactions with the amide bond of Val1 of **9**. We reasoned that if these residues are critical for binding of **9**, mutations to Ala should impair activity. Both Gln289A and Gln229A mutations were found to either result in complete loss of activity or impaired turnover of **9** and/or **13** (Fig. 5d, e). These data strongly support the binding of **9** in a U-shaped conformation. Moreover, a U-shaped conformation of **9** also nicely places the hydrophobic sidechains of **9** into hydrophobic pocket of the enzyme (Fig. 5b and Supplementary Fig. 23a) comprising residues that show varying degree of conservation (Fig. 5c, Supplementary Fig. 23b). Compound **9** is held in place primarily by hydrophic interactions such that it positions the carbonyl group (point of hydride attack) towards the $F_{420}$ and His62 to ensure the correct enantioselective outcome of the reaction (Fig. 5c and Supplementary Fig. 25; *vide infra*).

The ketone is optimally placed for proton/hydride transfer, with the carbonyl carbon placed 2.9 Å and 3.8 Å from the Nε of His62 and C5 of $F_{420}$, respectively. The side chain of Tyr122 packs against the Leu side chain and the β-keto group of PreSta3 of **9** (2.7 Å; Fig. 5c and Supplementary Fig. 23c). We thus wondered whether Tyr122 played a role in ketone reduction and produced kvPepI[Y122A] and kvPepI[Y122F]. When we tested the effect of kvPepI[Y122A] on **9**, we found ketone reduction to be significantly impaired, with only trace amounts of **13** observed (Fig. 5d). In contrast, kvPepI[Y122F] resulted in an accumulation of **13** (Fig. 5d). Moreover, we tested the activities of kvPepI[Y122A] and kvPepI[Y122F] in comparison to kvPepI using purified **13** and observed only a tiny amount of **17** with kvPepI[Y122F] but no formation of **17** was observed with kvPepI[Y122A] (Fig. 5e). This implies that Tyr122 is crucial and acts at two points during catalysis: a) binding of **9**, with hydrophobic interaction between phenyl ring and Sta helping orient the substrate for catalysis (Fig. 5c and Supplementary Fig. 23a, c), and b)

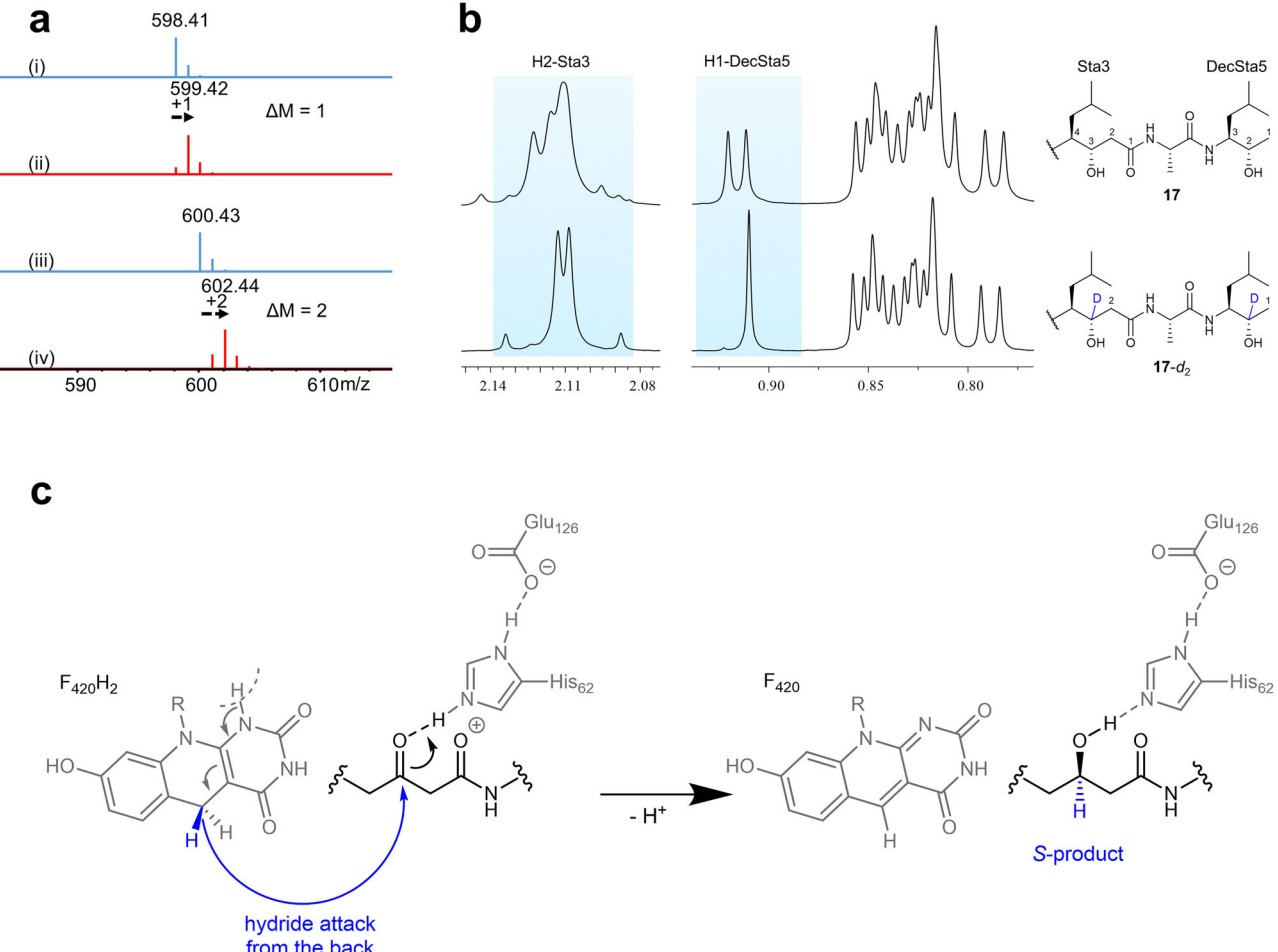

**Fig. 6 | Deuterium-labeled assays and mechanistic considerations for PepI. a** MS analysis of **13** and **17** in the kvPepI and **9** reactions without (i: **13**, iii: **17**, blue) or with (ii: **13**-$d_1$, iv: **17**-$d_2$, red) $F_{420}$-5-$d_1$ provided by the coupled hexokinase assay using D-Glucose-1-$d_1$. **b** The 2 Da increase in the molecular mass indicated **17**-$d_2$ was the deuterated derivative of **17**. The position of deuteration was deduced by comparing the ${}^1$H-NMR spectra. The methyl group H1-DecSta5 in **17**-$d_2$ showed a singlet whereas in **17** it was a doublet, together with the disappearance of the H2-DecSta5 signal in

**17**-$d_2$, indicating the H2-DecSta5 was deuterated. Similarly, the H3-Sta3 was deuterated, as evidenced by the disappearance of H3-Sta3 signal and splitting pattern change of H2-Sta3. **c** Proposed reaction mechanism derived from the experimental observations indicates the orientation of the pepstatin peptide backbone in relation to His62 and $F_{420}$ to achieve the observed *S*-product. R represents the lacty-loligoglutamate tail of $F_{420}$.

during the second round of reduction by forming hydrogen bonding with **13**. This likely involves the reduced ketone as kvPepI$^{Y122F}$ and kvPepI$^{Y122A}$ compared to kvPepI, either resulting in complete or significant loss of activity upon incubation with **13** (Fig. 5e). A comparison of the structure of kvPepI$^{Y122A}$-$F_{420}$ complex to the corresponding structure of the wild-type holoenzyme revealed no major structural perturbations caused by the mutation (Supplementary Fig. 23d and Supplementary Table 7).

Whether the kvPeI employs either processive[43] or distributive catalysis[44] is not clear. The accumulation of **13** during time-course experiments (Fig. 4e) hints towards the latter. The conformational changes required to position the second keto reduction site close to His62 could be facilitated by the flexible loop located above the Sta (Fig. 5b; magenta). The first ketone reduction may also allow the formation of intra- and intermolecular hydrogen bonding, assisting reorientation of **13**. Therefore, we envision a scenario where all the aforementioned factors likely contribute to the conformational change(s) needed to position the second keto group close enough to His62 for hydride transfer. Despite persistent efforts, we encountered challenges in obtaining the structure of kvPepI or variants in complex with $F_{420}$, **13** or **17**. Therefore, further studies are needed to gain a deeper

understanding of the binding mode and molecular mechanism of the second keto reduction.

## Hydride transfer from $F_{420}H_2$ to the keto carbon

Since we observe only the *S*-enantiomer of **9**, the specific orientation of substrate towards the $F_{420}$ and His62 must be decisive in the enantioselective outcome of the reaction. To shed light on how and where the hydride is transferred from the C5 atom of $F_{420}H_2$, we performed kvPepI assays with deuterium-labeled reduced $F_{420}$-5-$d_1$. This labeled cofactor was synthesized through a coupled reaction using D-Glucose-1-$d_1$ and hexokinase. Intermediates **13** and **17** were successfully produced and the main mass isotope ion increased by 1 and 2 Da, respectively (Fig. 6a). The mass shift indicates the incorporation of one deuterium in **13** and two deuteriums in **17**. To determine the exact position of incorporation, we purified the labeled **13** and **17** from a scaled-up reaction. The NMR analysis unambiguously confirmed that the deuterium is attached to the carbon atom bearing the hydroxyl group (reduced from the ketone) (Fig. 6b, Supplementary Figs. 73–74 and Supplementary Table 10). These findings demonstrate that a direct hydride shift takes place from the cofactor to the carbonyl carbon. Furthermore, the stereo-configuration (*S*) of the carbon attached to

the hydroxyl group in the middle Sta residue provides insight into the orientation of the pepstatin peptide backbone within the active site. In order to achieve the observed *S*-configuration while enabling protonation from His62 (*vide supra*), the peptide chain needs to be oriented as shown in Figs. 5c and 6c.

## Discussion

Pepstatins have gained significant attention due to their potent inhibitory effects against various APs, which are promising therapeutic targets for treating a number of important diseases[7]. Despite this interest, the biosynthesis pathway of pepstatins has remained unknown. In this study, we identified the NRPS-PKS hybrid BGC responsible for pepstatin biosynthesis and revealed its distinctive disconnected and iterative utilization pattern. Most notably, we uncovered a unique $F_{420}H_2$-dependent post-assembly modification mechanism, which plays a crucial role in constructing the essential 3-OH-4-NH$_2$ framework. This mechanism differs significantly from the previously hypothesized PKS KR-dependent pathway observed in the biosynthesis of other natural products (Fig. 1c)[26].

The concise pepstatin assembly line stands out due to its highly disconnected nature, featuring unusual *trans*-acting and iterative enzymes. The potential iterative use of the discrete A domain PepG and its in-trans cooperation with PepH and PepA warrant further in-depth investigation into the biosynthetic mechanism. Furthermore, the stand-alone NRPS/PKS modules (PepB, PepC, and PepD) involved in the assembly of the Sta-Ala-Sta chain may operate through mechanisms distinct from those proposed for thalassospiramide biosynthesis, which involves a cross-modular pass-back chain extension strategy within a single multimodular megaenzyme[39].

$F_{420}$-dependent oxidoreductases have been reported in several natural product biosyntheses, typically mediating the hydrogenation of different double bonds in alkene, ketone, and imine moieties[45–53]. Notably, PepI stands out as the only biochemically characterized $F_{420}$-dependent oxidoreductase capable of performing two rounds of reductions, underscoring the iterative fashion of pepstatin biosynthesis. Deletion of *pepI* led to the accumulation of β-diketone intermediates and the loss of pepstatin production. Through comprehensive structural and biochemical characterization of PepI/kvPepI, we confirmed that PepI/kvPepI sequentially reduces the middle β-ketone followed by the *C*-terminal β-ketone or methyl ketone.

Site-directed mutagenesis and kvPepI-**9** modeling studies support the role of His62 as a potential catalytic residue, as corroborated by the abolishment of enzymatic activity regardless of whether **9** or **13** were used as the substrate. Two additional residues, Tyr122 and Gln229, were also found to be critical for PepI activity and, based on the modeling studies, are likely to be involved in substrate binding. Interestingly, mutating Tyr122 to Phe had a much greater effect on the second ketone reduction compared to the first step reduction, suggesting the vital role of the hydroxyl group of Tyr122 in positioning the substrate for the second reduction. In addition, the loss of kvPepI activity in the Q229A mutant supports the crucial role of Gln229 in stabilizing the substrate binding by forming hydrogen bonding in the modeling prediction. Interestingly, structural tolerance of the *C*-terminal ketone reduction indicates that the diketone is not a prerequisite for PepI activity. Further studies are currently underway to understand how **13** is oriented within the active site during the second reduction step.

In conclusion, this study unveils the long-sought-for, highly dissociated, and iterative biosynthetic pathway of pepstatins. The discovery of the KR domain-independent 3-OH-4-NH$_2$ framework biosynthesis and the characterization of PepI as a tandem $F_{420}H_2$-dependent oxidoreductase expand the NRPS/PKS biosynthetic repertoire. Moreover, the data presented here on the PepI mechanism will likely contribute to a better understanding of underinvestigated $F_{420}$-dependent oxidoreductases.

## Methods

### Bacterial strains, plasmids, and DNA manipulation

Bacterial strains and plasmids used in this study are listed in Supplementary Tables 1 and 2, respectively. All primers used in this study (listed in Supplementary Table 3) were synthesized by Sigma-Aldrich (Steinheim, Germany). In-silico analyses, including primers design, BGC annotation, and short-read mapping, were conducted using Geneious Prime® 2022.2.2 (Biomatters Ltd., New Zealand). *E. coli* DH10B was employed for general subcloning purposes. Plasmid DNA isolation kits were obtained from Qiagen. Q5® High-Fidelity DNA polymerase and HiFi DNA Assembly Mater Mix were procured from New England Biolabs (Ipswich, MA). Restriction enzymes and other standard molecular biology reagents were purchased from Thermo Scientific. All kits and enzymes were used in accordance with the manufacturers' protocols. Red/ET recombineering–based plasmid modifications were performed using *E. coli* GB08-red, as described in previous studies[54,55]. Genomic DNA of *Streptomyces* strains was manually extracted following a cetyltrimethylammonium bromide-based protocol[56].

### Strain cultivation

*E. coli* was cultivated in LB medium (10 g/L tryptone, 5 g/L NaCl, 5 g/L yeast extract, pH 7.6) at 37 °C with shaking at 180 rpm. All *Streptomyces* strains were cultivated at 30 °C with shaking at 200 rpm. For *Streptomyces catenulae* DSM40258 and heterologous expression mutants, cells were grown in tryptone soya broth (TSB) medium (30 g/L TSB) for genomic DNA isolation, conjugation, or as starter cultures prior to fermentation. Conjugations were performed on MS agar plates (D-mannitol 20 g/L, soya bean meal 20 g/L, agar 20 g/L). *Streptomyces catenulae* medium (30 g/L glucose, 20 g/L corn steep liquor, 0.5 g/L dipotassium hydrogen phosphate, 0.2 g/L ammonium sulfate, 0.5 g/L magnesium sulfate, 5 g/L calcium carbonate, pH 7.0)[57] was used for fermentation. Fermentation cultures were inoculated from 3 to 4 days TSB starting cultures (10% (v/v) inoculation volume) and cultivated for 4 days (24 h for Del14-*pep*-Δ*pepI*).

### Genome sequencing and analysis

The genome of *Streptomyces catenulae* was sequenced using Illumina HiSeq and MinION Nanopore sequencing technologies at the genome analytics facility of the Helmholtz Centre for Infection Research (Braunschweig, Germany). BGCs were predicted by antiSMASH (http://antismash.secondarymetabolites.org/)[58].

### Construction of gene *pepD* deletion plasmid

The *pepD* gene in the chromosome of *Streptomyces catenulae* was deleted using the CRISPR-Cas9 gene editing tool pQS9, following previously described methods[59]. Spacer inserts, which included a gene-specific 20-nt guide sequence, were generated by annealing two 34-nt synthesized oligonucleotides, sgpepD-f and sgpepD-r. The annealed oligonucleotides were cloned into pQS9 at *Nco*I-*Xba*I to afford psgQS9-Δ*pepD* and verified by Sanger sequencing. The flanking region of gene *pepD*, 2 kb on each side, served as homologous recombination repair templates following Cas9-mediated gene editing. The 2.0 kb upstream homologous arm (UHA) and 2.0 kb downstream homologous arm (DHA) were amplified from genomic DNA of *Streptomyces catenulae* using primer pairs pepD-L-f/pepD-L-r and pepD-R-f/ pepD-R-r, respectively. These amplified fragments were cloned into psgQS9-Δ*pepD* at *Stu*I using the Gibson assembly cloning kit, resulting in the plasmid pQS9-Δ*pepD*, which was verified by *Sal*I digestion.

### Heterologous expression of the *pep* BGC

The potential pepstatin BGC *pep* was divided into three fragments, each ~6 kb in size, with 20 bp homologous arms flanking both ends. These fragments were amplified using the primer pairs pep-1-f/pep-1-r,

pep-2-f/pep-2-r, and pep-3-f/pep-3-r, respectively. Concurrently, the backbone plasmid was amplified using the primer pair p15A-f/p15A-r. The amplified plasmid backbone and the three *pep* gene cluster fragments were assembled via Gibson assembly, resulting in the formation of the plasmid p15A-*pep*.

Subsequently, a 4.3 kb phi31 integrase-apramycin resistance gene cassette was excised from the pR6K-phiC31-oriT plasmid through *AseI* digestion. This cassette was then introduced into the p15A-*pep* plasmid via Red/ET recombineering, yielding the recombinant plasmid p15A-int-*pep*. Plasmid p15A-int-*pep* was verified by *NcoI* digestion.

### Pathway engineering of the *pep* BGC for heterologous expression

The gene *pepJ* in p15A-int-*pep* was replaced by the chloramphenicol resistance gene (*cml*), which was amplified using the primers pepJ-ko-f and pepJ-ko-r, via Red/ET recombination[2,54]. This process yielded the plasmid p15A-int-*pep*-Δ*pepJ*.

To enhance pepstatin production in the heterologous expression mutant, the upstream non-coding region of gene *pepJ* in plasmid p15A-int-*pep* was substituted with a *cml-kasO* cassette using Red/ET recombination, resulting in the plasmid p15A-int-*pep*-kasop-*pepJ*[55].

For *pepI* inactivation, both *pepI* and the upstream non-coding region of *pepJ* in plasmid p15A-int-*pep* were replaced with *cml-kasO* cassette via Red/ET recombination, producing the plasmid p15A-int-*pep*-Δ*pepI*[55].

### Conjugation

In this study, we employed an adapted biparental or triparental conjugation protocol[56] to introduce exogenous DNA into *S. catenulae* or *S. albus* Del14, respectively. The donor strain used for biparental conjugation was *E. coli* ET12567/pUZ8002. For triparental conjugation, the donor strain was *E. coli* DH10B, with *E. coli* HB101/pRK2013 serving as the helper strain.

After conjugation, the plates were allowed to dry and then further incubated at 30 °C until the exconjugants became visible, typically within 3–5 days. Visible exconjugants were inoculated to TSB media for genome DNA isolation. Each exconjugant was verified by PCR amplification with relevant primers (Supplementary Table 3).

### Gene cloning and protein purification

The codon-optimized *pepI* gene was synthesized by GenScript and subsequently amplified using the primers pepI-f/pepI-r. The amplified *pepI* was cloned into the plasmid pCold I, resulting in the plasmid pCold-pepI. Additionally, the plasmids pET28b-sfpepI, pET28b-svpepI, and pET28b-kvpepI were ordered from GenScript.

For expressing the kvPepI site-directed mutants, H62A, Y122F, Y122A, and Q229A, plasmid pET28b was digested with *NdeI*/*Hind*III for cloning. The *kvpepI* gene was split into two parts, and the specific mutations were introduced at the junction through primers listed in Supplementary Table 3. The resulting mutant fragments were assembled into the plasmid pET28b via Gibson assembly. For the Q289A mutant, the pET28b plasmid was amplified using the primer pair pET-28b-f/pET-28b-r to serve as the backbone. The mutation was introduced through primers (listed in Supplementary Table 3) at the junction and cloned to plasmid pET28b by Gibson assembly.

All expression constructs were transformed into BL21 (DE3) cells. For protein purification, cells were grown overnight in LB containing relevant selection pressure at 37 °C. This overnight culture was diluted 1:100 into fresh LB medium supplemented with the appropriate antibiotics and incubated at 37 °C with shaking for 3–4 h. Until optical density (OD$_{600}$) reached 0.6, the culture was rapidly cooled to 16 °C in ice water for 30 min. Subsequently, isopropyl β-ᴅ-1-thiogalactopyranoside (IPTG) was added to a final concentration of 0.1 mM to induce protein expression. The culture was then incubated with shaking at 16 °C for 16 h. Following this incubation period, the cells were harvested by centrifugation.

For protein purification, cell pellets were resuspended in lysis buffer (20 mM Tris pH 8.0, 200 mM NaCl, 10% glycerol (w/v), 20 mM imidazole, 1 mM TCEP) supplemented with cOmplete EDTA-free protease inhibitor tablets (Roche). The cell suspension was lysed by two passages through a cell disrupter (30 kpsi, Microfluidics Corp.), and the cell debris was removed by centrifugation at 40,000 × g for 30 min at 4 °C. The supernatant was then loaded onto a 5 mL HisTrap HP column (GE Healthcare) pre-equilibrated with the lysis buffer. The column was washed with 30 column volumes of lysis buffer, and the bound protein was eluted using an elution buffer (lysis buffer with 250 mM imidazole).

Fractions containing the target protein were subjected to size-exclusion chromatography using a HiLoad 16/600 Superdex 200 pg column (GE Healthcare) pre-equilibrated with gel filtration buffer (50 mM Tris pH 8.0, 200 mM NaCl, 1 mM TCEP). Protein purity was assessed by SDS-PAGE, and the fractions containing the highest purity protein were pooled and concentrated to approximately 5 mg/mL. The protein concentration was determined using a Nanodrop UV-Vis spectrophotometer at 280 nm, with theoretical extinction coefficients calculated using the ExPASy ProtParam tool.

### Enzyme assay conditions

The crude extract containing compounds **5-8** previously stored at −80 °C, was dissolved in pre-cooled methanol to a concentration of 1 mg/mL and stored on ice. Purified compounds **9-12** were separately dissolved in methanol to create 1 mM stock solutions. The F$_{420}$ and FGD enzyme used in the activity assays were prepared as previously described[60,61].

A 50 μL reaction mixture was prepared in 1.5 mL Eppendorf tubes, containing 0.6 μM PepI, kvPepI, or other mutants, 20 mM Tris-HCl (pH 7.5), 10 μM F$_{420}$, 2.5 mM glucose 6-phosphate, 0.45 μM F$_{420}$-dependent glucose-6-phosphate dehydrogenase (FGD) and 10 μM substrate (or 2 μg crude extract). In the time-course experiments of kvPepI, substrate concentration was reduced to 1 μM. Unless otherwise noted, the reaction mixture was incubated at 30 °C for 5 h, then quenched with 0.2 M HCl, and stored at −80 °C until ready for analysis.

Before LC-MS analysis, an equal volume of methanol was added to the reaction mixture, and it was centrifuged at 8000 × g for 15 min at 15 °C to remove all the particles.

### Sample preparation and UPLC-ESI-MS analysis

For small-scale fermentation (50 mL), *Streptomyces* cultures were harvested by centrifugation at 8000 × g for 15 min at 15 °C. The supernatant was supplemented with 2% (v/v) XAD16N resin and stirred for 2 h, followed by extraction with 50 mL methanol. The pelleted cells were resuspended in 50 mL methanol and agitated for 2 h. All fractions were evaporated to dryness under vacuum and then dissolved in 1 mL methanol to produce the crude extracts. The crude extract (10 μL) was diluted fivefold to 50 μL and centrifuged at 21,500 g for 15 min at 15 °C before UPLC-MS analysis.

Reverse phase UPLC-MS analysis was carried out using the following system and method. LC: Ultimate 3000 RS; HRMS: Bruker Maxis II (4Generation) Q-TOF using the Apollo II ESI source.; MS: Bruker amaZon speed ion trap mass spectrometer; Column: ACQUITY UPLC BEH C18 Column, 130 Å, 1.7 μm, 2.1 mm × 50 mm (Waters); Eluents: A: distilled water supplemented with 0.1% formic acid and B: distilled acetonitrile supplemented with 0.1% formic acid; Flow rate: 0.6 mL/min; column temperature: 45 °C. The gradient was as follows: (1) a 0.5 min isocratic step at 5% B, changed from 5% to 95 % B in 9 min, maintained at 95 % B for 1 min, decreased to 5% B in 0.5 min, and sustained 5% B for 2 min. (2) a 0.5 min isocratic step at 5% B, followed by a linear increase to 95% B in 18 min, maintained at 95 % B for 2 min, then equilibrated to the starting conditions (5% B) for 2 min. The UV-

Vis spectra were recorded by a diode array detection (DAD) in the range from 200 to 600 nm. The mass detection was performed in the positive ESI mode. All HPLC-MS data were analyzed by *Compass Data Analysis* version 4.4 (Bruker Daltonics).

## Compounds purification

To purify compounds **1-4**, the cells and XAD16N resin from a 3-liter *S. catenulae* culture were harvested by centrifugation. The combined cells and resin were extracted with methanol. The extracts were concentrated using a rotary evaporator and then partitioned between methanol and hexane. The methanol phase was dried and further purified on a Sephadex LH-20 column (GE Healthcare) with methanol as the mobile phase. Fractions were pooled based on HPLC-MS analysis and were further purified by HPLC.

Semi-preparative purification of pepstatins was performed on an HCT HPLC-MS system using a Waters XSelect Peptide CSH C18 column (5 μm, 10 × 250 mm). The HPLC conditions were as follows: solvent A, $H_2O$ ( + 0.1% formic acid), and solvent B, ACN ( + 0.1% formic acid), at a flow rate of 5 mL/min and a column thermostatic at 45 °C. The gradient starts with a 2-minute isocratic step at 25 %B, followed by a ramp to 53% B in 16 min, maintained at 95%B for 3 min before returning to the initial condition in 2 min and re-equilibration for 2 min.

To purify compounds **9-12**, a 3-liter culture of *S. albus* Del14-pep-Δ*pepI* was grown in liquid medium for 2 days, followed by the addition of activated carbon (1%, w/v). After centrifugation, the supernatant was removed, and the cells with activated carbon were extracted with methanol. The extracts were dried, and partitioned between ethyl acetate and deionized water (1:1, v/v). The organic phase containing compounds **9-12** was concentrated by rotary evaporation, and subsequently fractionated using flash chromatography on a Biotage Isolera One system with a 25 g Snapfit column. Elution was performed with ethyl acetate and methanol in the following gradient: 5 CV (column volume) of ethyl acetate, followed by a 10 CV linear increase from 0 to 20% methanol, and then a 10 CV linear increase from 20 to 100% methanol. Fractions were collected and further purified using a Dionex Ultimate 3000 SDLC low-pressure gradient system with a Waters XSelect Peptide CSH C18 column (5 μm, 10 × 250 mm). The gradient conditions were as follows: 0−2 min, 20%B; 2−12 min, 20−50%B; 12−15 min, 50-56%B; 15−17 min, 56−95%B; 17−19 min, 95%B; 19−-21 min, 95-20%B; 21−23 min, 20%B, with $H_2O$ ( + 0.1% formic acid) as eluent A and ACN ( + 0.1% formic acid) as eluent B. The separation was carried out with a column temperature of 45 °C at a flow rate of 5 mL/min.

To purify compounds **13** and **17**, an in vitro ketoreduction reaction of substrate **9** was performed using the enzyme kvPepI, scaled up proportionally to 600 μL according to the method described in enzyme assay conditions. The reactions were quenched and extracted with ethyl acetate. The extracts were then concentrated and purified by HPLC, using the same separation conditions as those for intermediates **9-12**.

## Statine residue conformation verification

For ʟ-statine (AAT Bioquest): Prepare two 1.5 ml Eppendorf tubes and add 50 μl of 10 mM ʟ-statine solution to each. Adjust pH to approximately 9 by adding 20 μl of 1 M NaHCO₃ to each tube. Then, add 20 μl of 1% Marfey's reagent[62] in acetone (ᴅ-FDLA and ʟ-FDLA, respectively) to each tube, and incubate at 40 °C with shaking at 700 rpm for 1 to 2 h. Subsequently, neutralize the reaction by adding 10 μl of 2 N HCl, then dilute with 300 μl of acetonitrile. Centrifuge the mixture at 8000 × *g* for 15 min at 15 °C before UPLC-MS measurement. An aliquot (1 μL) of the analyte was injected, and the separation was achieved on a Waters Acquity BEH C18, 100 × 2.1 mm, 1.7 μm column using $H_2O$ ( + 0.1 % FA) and ACN ( + 0.1 % FA) as eluents. The flow rate is 550 μL/min at a column temperature of 45 °C.

For compounds **9, 13, 17**, and **1**, samples (approximately 100 μg each) were placed in 1.4 ml glass vials and heated in 100 μl of 6 M HCl at 110 °C for 45 min. The solvent was evaporated using a nitrogen flow at room temperature, and the residues were dissolved in 110 μL of $H_2O$. Derivatization and analysis were performed in the same way as described above.

## NMR conditions

NMR data were recorded in methanol-$d_4$ or DMSO-$d_6$ on a 500 MHz Avance III (UltraShield) spectrometer or a 700 MHz Avance III (Ascend) spectrometer, each equipped with a Helium-cooled CryoProbe (TCI). All observed chemical shift values ($\delta$) are given in ppm and coupling constant values (*J*) in Hz. Chemical shifts were calibrated internally to the residual signal of either methanol-$d_4$ ($\delta_H$ 3.31, $\delta_C$ 49.1) or DMSO-$d_6$ ($\delta_H$ 2.50, $\delta_C$ 39.5). Multiplicities are described using the following abbreviations: s = singlet, d = doublet, t = triplet, q = quartet, dd = doublet of doublets, m = multiplet, br = broad signal.

## Compound characterization

Acetyl-pepstatin (**1**): the molecular formula of **1** was determined to be $C_{31}H_{57}N_5O_9$ by High-resolution ESI-MS, *m/z* 644.4230 [M + H]⁺ (calcd. for $C_{31}H_{58}N_5O_9$, 644.4229). Analysis of the 1D and 2D NMR spectra (Supplementary Figs. 26−29, Supplementary Table 8), as well as the MS/MS fragmentation pattern (Supplementary Fig. 1), revealed that **1** has a peptide sequence of Ac-Val-Val-Sta-Ala-Sta. Hence, **1** was determined to be acetyl-pepstatin.

Propionyl-pepstatin (**2**) had the molecular formula $C_{32}H_{59}N_5O_9$, *m/z* 658.4383 [M + H]⁺ (calcd. for $C_{31}H_{60}N_5O_9$, 658.4386). The 1D NMR spectra of **2** were almost identical to that of **1**, except for an additional $CH_2$ group. Further interpretation of the 2D NMR data revealed a propionylated N-terminus. Thus, **2** was identified as propionyl-pepstatin (Supplementary Figs. 30−33, Supplementary Table 8).

Isobutyryl-pepstatin (Pepsinostreptin A) (**3**) had the molecular formula $C_{33}H_{61}N_5O_9$, *m/z* 672.4518 [M + H]⁺ (calcd. for $C_{33}H_{62}N_5O_9$, 672.4542). The 1D and 2D NMR spectra of **3** were similar to that of **1**, the only difference was the N-terminal isobutyl group instead of the singlet methyl group. The structure of **3** was determined as isobutyryl-pepstatin (Supplementary Figs. 34−38, Supplementary Table 8).

2-methylbutanoyl-pepstatin (**4**) had the molecular formula $C_{34}H_{63}N_5O_9$, *m/z* 686.4660 [M + H]⁺ (calcd. for $C_{34}H_{64}N_5O_9$, 686.4699). The NMR data (Supplementary Figs. 39−42, Supplementary Table 8) and MS/MS fragmentation data (Supplementary Fig. 1) showed that **4** has the same amino acid sequence as **1-3**. Thorough examination of the NMR data revealed that the fatty acid chain of **4** consisted of one methine group ($\delta_{H-2}$ 2.36, $\delta_{C-2}$ 43.4), one methylene group ($\delta_{H-3}$ 1.38/1.61, $\delta_{C-3}$ 28.3), and two methyl groups ($\delta_{H-4}$ 0.87, $\delta_{C-4}$ 12.4; $\delta_{H-5}$ 1.11, $\delta_{C-5}$ 18.2), indicating a 2-methylbutyl moiety. Therefore, the structure of **4** was characterized as 2-methylbutanoyl-pepstatin.

The molecular formula of compound **9** was determined to be $C_{30}H_{53}N_5O_7$ by ESI-HRMS, *m/z* 596.4017 [M + H]⁺ (calcd. for $C_{30}H_{54}N_5O_7$, 596.4018). 1D and 2D NMR data (Supplementary Figs. 43−47, Supplementary Table 9) in combination with mass spectrometric analysis (Supplementary Fig. 10) showed the presence of an oxidized statin residue (PreSta), with the hydroxy group converted into ketone group ($\delta_{C-3}$ 204.9). In addition, a decarboxylated and oxidized statin residue (DecPreSta) was identified. Finally, two valine, one alanine, and one acetyl group were identified. HMBC correlations together with MS/MS analysis determined the amino acid sequence, hence the planar structure of **9** was characterized.

Similarly, careful analysis of the NMR spectra (Supplementary Figs. 48−62, Supplementary Table 9) and MS/MS data (Supplementary Fig. 10) of compounds **10-12** showed they are the decarboxylated and oxidized intermedium of **2-3**, respectively. Hence, the structures of compounds **10-12** were determined.

The molecular formula of **13** was determined to be $C_{30}H_{55}N_5O_7$ by High-resolution ESI-MS, *m/z* 598.4179 [M + H]⁺ (calcd. for $C_{30}H_{56}N_5O_7$, 598.4174). The NMR data (Supplementary Figs. 63−67, Supplementary

Table 10) and MS/MS fragmentation data (Supplementary Fig. 16) indicated that the structure of **13** was similar to that of **9**. The only difference is the PreSta residue in **9** was replaced by Sta residue in **13**, indicated by the disappearance of one ketone signal ($\delta_{C-3}$ 204.9) and the appearance of one methine group ($\delta_{H-3}$ 3.84, $\delta_{C-3}$ 68.7).

Compound **17** had the molecular formula $C_{30}H_{57}N_5O_7$, *m/z* 600.4332 $[M + H]^+$ (calcd. for $C_{30}H_{58}N_5O_7$, 600.4331). Compare the 1D and 2D NMR spectra (Supplementary Figs. 68−72, Supplementary Table 10) of **17** and **13** revealed that the ketone signal ($\delta_{C-2}$ 207.9) in **13** was replaced by the methine group ($\delta_{H-2}$ 3.56, $\delta_{C-2}$ 67.2) in **17**. The structure of **17** was determined.

### Crystallization and structure determination of kvPepI
Crystals of kvPepI, kvPepI – $F_{420}$, kvPepI$^{H62A}$, kvPepI$^{H62A}$ – $F_{420}$, and kvPepI$^{Y122A}$ – $F_{420}$ were obtained at 18 °C in 30% (w/v) PEG 4000, 0.2 M Sodium acetate and 0.1 M Tris-Cl pH 8.5. For complex crystallization, the protein was incubated with excess $F_{420}$ (1.5 mM) on ice overnight. Crystals were cryoprotected in mother liquor supplemented with 30% glycerol. The diffraction data was collected from a single crystal at 100 K at Petra III (Beamline: P11, DESY)[63,64], processed using Xia2[65] or XDS[66], and the structure was determined using PHASER molecular replacement[67] using AlphaFold model generated using Colab notebook[68,69]. The structure was manually rebuilt in COOT[70], refined using PHENIX Refine[67], and validated using MolProbity[71]. The images presented were created using PyMOL (Schrödinger) and LigPlot$^{+72}$.

### Modeling of PepI: compound 9 complex
A model of the complex between the biosynthetic precursor of pepstatin and substrate of PepI (compound **9**) and the kvPepI protein was generated based on the solved crystal structure (*vide supra*, PDB ID 9GM0) after the unresolved flexible loop Gly191-Thr194 was modeled by AlphaFold[68,69] using the molecular operating environment (MOE, chemical computing group)[73] in a stepwise manner:

1. The PDB file was loaded in MOE and the QuickPrep protein preparation procedure was applied with the following parameters: Tether strength for receptor = 30; Tether strength for ligand = 10; co-factor ($F_{420}$) atoms were fixed; all other parameters were default.
2. The structure of compound **9** was inserted into the active site by first building the β-keto amide motif (precursor of the statine motif, "PreSta3") in the orientation where the carbonyl oxygen is occupying the position of His62-coordinated water ($H_2O570$; ordered water molecule from the main text) and the α-carbon, which is attacked by the hydride (or deuteride) in close proximity to the reactive center of co-factor $F_{420}$ in the only possible orientation, which would lead to generation of the observed enantiomer (*S*-product; see Supplementary Fig. 25).
3. Growing of the peptidic structure of compound **9** into the pocket and orienting the hydrophobic side chains into hydrophobic pockets. The Steric constraints of the active site required to model a compound **9** in a loop/bent conformation with the loop occurring between Val1 and PreSta3 to remove clashes with the protein resulting in a U-shaped conformation.
4. Removal of all clashing water or solvent molecules as well as further finetuning of the complex structure.
5. A final QuickPrep step for localized energy minimization with the following parameters: Tether strength for receptor = 30; Tether strength for ligand = 10; co-factor ($F_{420}$) atoms were fixed; all other parameters were default.

This yielded the proposed model for the interaction between PepI and the pepstatin precursor **9**. Note that this model is an information-driven hypothesis, which shows a plausible orientation for enabling the generation of the S-product at residue 4 observed in the experimental studies highlighting that this enzyme-substrate complex is in principle possible and that the active site can accommodate compound **9** in this orientation. In general, coordinates for substrate atoms more distant to the β-keto statine precursor motif are characterized by ambiguities inherent to modeling efforts on highly flexible peptidic substrates.

### Reporting summary
Further information on research design is available in the Nature Portfolio Reporting Summary linked to this article.

## Data availability
The pepstatin biosynthetic gene cluster sequence in this study has been deposited in GenBank under accession number PP947771. The protein crystal structure data generated in this study have been deposited in the Protein Data Bank under PDB IDs 9G64, 9GKH, 9GM0, 9GNC, and 9GND. All data that support the findings of this study are available in the main text, supplementary information and from corresponding author(s) upon request. Source data are provided with this paper.

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

## Acknowledgements

This work is supported by the Helmholtz International Lab (InterLabs-0007). Research in Chengzhang Fu and Rolf Müller's laboratory is funded by the Bundesministerium für Bildung und Forschung (BMBF). The authors acknowledge DESY (Hamburg, Germany), a member of the Helmholtz Association HGF, for the provision of experimental facilities. Parts of this research were carried out at PETRAIII (beamline P11), and we would like to thank Johanna Hakanpää, Guillaume Pompidor and Helena Taberma for assistance in using the photon beamline. Beamtime was allocated for proposal (Xh-20010236). Ghader Bashiri is supported by a Health Research Council of New Zealand grant (Hercus Health Research Fellowship 17/058).

## Author contributions

C.F. conceived the study and analyzed the gene cluster. J.M. performed heterologous expression of BGC and in vitro biochemical studies. A.S. and H.Z. performed the protein crystallographic studies. H.Z. isolated the compounds and determined the chemical structures. G.B. purified $F_{420}$ and FGD. M.E. performed the modeling. C.F., J.M., H.Z., A.S., L.H., M.E., and R.M. analyzed the data. C.F. and A.S. wrote the paper with input from all authors. J.M., H.Z., and A.S. contributed equally to the manuscript. All authors read and approved the final manuscript.

## Funding

## Competing interests

The authors declare no competing interests.
