## [Transparent Peer Review file · Nature Communications]

Tandem ketone reduction in pepstatin biosynthesis reveals an F₄₂₀H₂-dependent statine pathway

Corresponding Author: Dr Chengzhang Fu

Version 0:

Reviewer comments:

Reviewer #1

(Remarks to the Author)

In this manuscript, Mo et al. present the discovery and characterization of an unconventional gene cluster governing the pepstatin biosynthesis characterized by a combination of nonribosomal peptide synthetase and polyketide synthase genes. This observation inspired the authors to assume and confirm an undescribed (trans-acting and iterative) catalytic nature, where Pepl, an F₄₂₀H₂-dependent oxidoreductase, was evidenced to catalyze the consecutive reduction of the β-ketone motifs in the central and C-terminal statine moieties, thereby being distinct from the previously hypothesized biosynthesis involving polyketide synthase ketoreductase domains. Using kvPepl, a homolog of Pepl, the authors performed the structural studies, site-directed mutagenesis, and deuterium-labeled enzyme assays to probe the mechanism of F₄₂₀H₂-dependent oxidoreductases and to identify critical residues. With such a set of experimentation, the research team claimed another pepstatin biosynthesis unique in using the only known iterative F₄₂₀H₂-dependent oxidoreductase. The findings are interesting and thus principally merits its appearance in Nature Communications. However, in these referees' opinion, a list of demerits or shortcomings should be addressed or improved, prior to being publicized, such as (but NOT limited to):

1. The authors failed to obtain the purified Pepl protein, and thus used kvPepl (sharing 53.3% sequence identity with Pepl) from another strain for the ensuing structural and mutational analysis. Frankly speaking, this is a weakness of the work since such a sequence identity does not guarantee the (higher) identity in their catalytic feature. At least, it is desired to add a phylogenetic tree of Pepl and kvPepl proteins as well as their relatives, which, I hope, could provide additional rationalization for the point.
2. In line 94, "Identification of the pepstatin BGC defying the colinearity rule", the authors are not clearly explained why directly knocking out gene pepD and pepJ abolished the production of pepstatins 1-4. The deletion and heterologous expression of other genes (pepA, B, C, E, F, G, H, and I) should be elaborated in detail, too.
3. UPLC-HRMS analysis of Del14-pep-ΔpepJ (iv) in Figure 2a seems to disagree in part with the claim that the production of 1-4 was abolished by pepJ deletion since at least 4 remained detectable. Furthermore, "Del14-pep-ΔpepJ (v)" (line 105) should be "Del14-pep-ΔpepJ (iv)", and the next entry (line 106) should be specified with "(v)".
4. The NMR spectra of compound 4 looked a bit dirty. Accordingly, Figure S34-S37 (1H-NMR, 13C-NMR, HSQC, and COSY spectra of compound 4) should be supplied with the full spectra so that readers can understand the residual amount of substrates with isolated keto groups that should resonate around 210 ppm.
5. Upon its first-time appearance, F₄₂₀ should be defined as done elsewhere (e.g., Journal of Bacteriology 1975, 121, 154-191) so that junior readers such as graduate students can follow the story easily.
6. Many expressions/mistakes should be polished/corrected. For example, the first two sentences are not unclear enough for readers to perceive the key part of the issue the work wishes to address. To list a few more, "forms hydrogen bond interactions with" should be changed into "forms hydrogen bondings with" (line 47); "the biosynthetic pathway responsible for pepstatin production" into "the pepstatin biosynthetic pathway" (line 52); "Identification of pepstatins and the related pep biosynthetic gene cluster" into "Identification of pepstatins and the pep cluster (pep in italic)" (line 102); and in line 183, "(Fig. 11)" should be "(Supplementary Figs.11)". The authors need to re-check each sentence while improving the manuscript.

Reviewer #2

(Remarks to the Author)

Mo et al describe the discovery of an unexpected enzyme that is essential for pepstatin biosynthesis. It is an oxidoreductase

that employs F420 for the reduction of two keto groups on the statin scaffold to generate the final product. This activity is appropriately described as "tailoring". The mechanistic properties of the enzyme are analyzed in-depth and follow the expected hydride transfer scheme. The interesting feature is that the enzyme performs two reductions following a defined order. The manuscript also discusses the overall biosynthetic pathway, predicting the enzymes involved and the reactions catalyzed. However, no experimental proof is provided in support of this hypothesis.

Uncovering a new step in pepstatin biosynthesis is welcome stepforward that adds to our knowledge and understanding about the biochemistry underlying such an important class of natural products. This will also clarify past literature. The work also expands our knowledge about F420 enzymes though the reaction catalyzed by pepI is not unusual for this class of oxidoreductases.

The manuscript is particularly strong with regards to the methods. It is a remarkable combination of microbial genetics, analytical chemistry, enzymology, and structural biology. The conclusions of the manuscript concerning the F420-dependent pepI therefore rely on and are corroborated by a solid experimental ground. The experimental evidence supporting the overall pathway for pepstatin biosynthesis is instead much more limited and essentially based on structural similarities. For instance, the activity of the A domain of pepD is based on "the similarity of threonine to valine" without any further element.

I have two main suggestions

-The statements about the overall pathway should be rephrased to avoid any overclaiming. For instance in the discussion ("we unravel the highly disconnected and iterative NRPS-PKS hybrid biosynthetic pathway for pepstatins...The concise pepstatin assembly line stands out due to its highly disconnected nature, featuring unusual trans-acting and iterative enzymes...") as well as in the introduction ("we shed light on a noncanonical NRPS-71 PKS BGC responsible for pepstatin biosynthesis featuring in-trans and iterative NRPS and PKS, deviating significantly from the so-called "colinearity" principle..."). The manuscript puts forward a sensible and likely correct hypothesis but there is almost no data supporting it if not for the tailoring role of pepI.

-Can the Authors comment about other compounds of the statin family? Are they synthesized through the same tailoring activity discovered for pepstatin? Figure 1a shows many compounds and figure 4b displays some genomic comparisons. Is this a pepstatin-specific biosynthetic enzyme or is it likely to be a general feature of statin biosynthesis?

Version 1:

Reviewer comments:

Reviewer #1

(Remarks to the Author)

In my personal opinion, two obvious shortcomings should be improved as specified below:

1. The NMR spectra of compound 4 looked dirty, and a set of pure sample based NMR spectra (¹H-NMR, ¹³C NMR, and 2D-NMR spectra) is desired.
2. The results regarding the deletion and heterologous expression of some other genes (pepA, B, C, E, F, G, H, and I) should be mentioned in the paper.

Reviewer #2

(Remarks to the Author)

The manuscript has been carefully and convincingly revised. Very good work.

Point-by-point response:

Reviewer 1

In this manuscript, Mo et al. present the discovery and characterization of an unconventional gene cluster governing the pepstatin biosynthesis characterized by a combination of nonribosomal peptide synthetase and polyketide synthase genes. This observation inspired the authors to assume and confirm an undescribed (trans-acting and iterative) catalytic nature, where PepI, an $F_{420}H_2$ -dependent oxidoreductase, was evidenced to catalyze the consecutive reduction of the β -ketone motifs in the central and C-terminal statine moieties, thereby being distinct from the previously hypothesized biosynthesis involving polyketide synthase ketoreductase domains. Using kvPepI, a homolog of PepI, the authors performed the structural studies, site-directed mutagenesis, and deuterium-labeled enzyme assays to probe the mechanism of $F_{420}H_2$ -dependent oxidoreductases and to identify critical residues. With such a set of experimentation, the research team claimed another pepstatin biosynthesis unique in using the only known iterative $F_{420}H_2$ -dependent oxidoreductase. The findings are interesting and thus principally merits its appearance in Nature Communications. However, in these referees' opinion, a list of demerits or shortcomings should be addressed or improved, prior to being publicized, such as (but NOT limited to):

Response: We sincerely appreciate your thoughtful evaluation and positive feedback on our manuscript. Your recognition of the novelty and significance of our findings, is highly encouraging. Below, we address the specific shortcomings raised and provide additional clarifications to strengthen the manuscript.

1. The authors failed to obtain the purified PepI protein, and thus used kvPepI (sharing 53.3% sequence identity with PepI) from another strain for the ensuing structural and mutational analysis. Frankly speaking, this is a weakness of the work since such a sequence identity does not guarantee the (higher) identity in their catalytic feature.

Response:

We would like to clarify that the functional characterization of authentic PepI has indeed been performed and is thoroughly described in our original manuscript. We successfully purified recombinant PepI, as evidenced by SDS-PAGE analysis (Supplementary Fig. 11b, Lane 4). Additionally, the functional characterization of PepI (Supplementary Figs. 13) demonstrated its activity as an $F_{420}H_2$ -dependent ketone reductase capable of reducing compounds **9-12** to generate compounds **13-16**.

We acknowledge that PepI exhibited limited solubility, which hindered its application in structural biology studies. To address this, we screened several homologs of PepI (Supplementary Fig. 11) and identified kvPepI as a suitable alternative due to its superior solubility. Functional assays confirmed that kvPepI retains $F_{420}H_2$ -dependent ketone reductase activity and exhibits functional similarity to PepI, strongly suggesting that both enzymes share similar catalytic features (Supplementary Figs. 14). Therefore, subsequent structural and mutational analyses were conducted using kvPepI as a surrogate.

To ensure transparency and prevent potential misunderstandings, we have revised the manuscript to clarify this point. Specifically, we have added the following text:

Page 7 and 8, Line 192-195:

"To investigate this novel mechanism of statine formation, we aimed to characterize PepI *in vitro*. Recombinant PepI was successfully purified from *E. coli* BL21 (DE3). However, the recombinant protein exhibited unsatisfactory solubility (Supplementary Fig. 11b), precluding its use in downstream structural biology studies."

At least, it is desired to add a phylogenetic tree of PepI and kvPepI proteins as well as their relatives, which, I hope, could provide additional rationalization for the point.

Response:

Thank you for the insightful suggestion. We have now included a phylogenetic tree of PepI, kvPepI, and their homologs in Supplementary Fig. 11a to provide additional context and further support for our findings. This addition helps to illustrate the evolutionary relationship between PepI, kvPepI, and related proteins, further rationalizing the use of kvPepI as a surrogate for structural and mutational studies.

We have also updated the main text to reflect this addition:

Page 8, Line 195-198:

"To address this challenge, we searched public databases for homologs of PepI and identified several *pep*-like BGCs harboring *pepI* gene homologs. These BGCs were classified into three distinct subtypes, primarily based on variations in their PKS genes (Fig. 4b, Supplementary Fig. 11a, and Supplementary Table 6)." We hope this addition enhances the clarity and scientific rigor of the manuscript.

2. In line 94, "Identification of the pepstatin BGC defying the colinearity rule", the authors are not clearly explained why directly knocking out gene *pepD* and *pepJ* abolished the production of pepstatins **1-4**. The deletion and heterologous expression of other genes (*pepA*, *B*, *C*, *E*, *F*, *G*, *H*, and *I*) should be elaborated in detail, too.

Response:

We sincerely appreciate your valuable comments. Since the *pep* BGC exhibits an unconventional NRPS-PKS architecture, we conducted targeted gene knockout experiments to confirm its involvement in pepstatin biosynthesis. Specifically, we decided to delete the NRPS gene *pepD*, which is supposed to be essential for the assembly line to construct the peptide backbone. Deletion of *pepD* completely abolished the production of pepstatins **1-4**, providing strong evidence that this BGC governs pepstatin biosynthesis. Additionally, heterologous expression of the entire *pep* BGC successfully led to pepstatin production, further verifying this relationship.

The deletion of other genes (*pepA*, *B*, *C*, *E*, *F*, *G*, and *H*) was not performed, as the deletion results of *pepD*, *pepI*, and *pepJ*, combined with the heterologous expression of the *pep* BGC, were sufficient to establish a definitive link between pepstatin and the *pep* BGC. Notably, *pepJ*, which encodes a LuxR-like transcriptional regulator, was identified as a positive regulator of pepstatin production. Deleting *pepJ* significantly reduced pepstatin levels, whereas its overexpression under the *kasOp* promoter increased production by more than 25-fold. This enhanced yield was crucial for downstream biosynthetic studies, such as isolating intermediate compounds following *pepI* deletion.

Additionally, we acknowledge the inaccurate use of the term "cessation" in our original text. Our intended meaning was "significant reduction." Accordingly, we have revised "led to cessation" to "resulted in a significant reduction" to more accurately reflect our findings. To further clarify this, we have included a direct comparison of the extracted ion chromatograms (EICs) of pepstatins from different strains in Supplementary Fig. 7a. This comparison clearly demonstrates the significant reduction in pepstatin production following *pepJ* deletion.

To address your concerns, we have revised the manuscript to clarify these points:

Page 5, Line 121-125:

"To confirm the involvement of the *pep* BGC in pepstatin biosynthesis, we performed gene deletion of *pepD*, which encodes the potential alanine assembly module. Deleting the NRPS gene *pepD* completely abolished pepstatin production in *S. catenulae* DSM40258 (Fig. 2a; Supplementary Fig. 3 and 7a), demonstrating that *pepD* is essential for pepstatin biosynthesis."

Page 5, Line 129-133:

"Deletion of the LuxR-like transcriptional regulator gene *pepJ* significantly reduced pepstatin production, suggesting that *pepJ* functions as a positive regulator. Conversely, overexpression of *pepJ* under the *kasOp* promoter led to a more than 25-fold increase in pepstatin yield (Fig. 2a and Supplementary Figs. 5–7), enabling the identification of biosynthetic intermediates in subsequent studies."

We hope this clarification adequately addresses your concerns and strengthens the manuscript.

3. UPLC-HRMS analysis of Del14-*pep*- Δ *pepJ* (iv) in Figure 2a seems to disagree in part with the claim that the production of 1-4 was abolished by *pepJ* deletion since at least 4 remained detectable. Furthermore, "Del14-*pep*- Δ *pepJ* (v)" (line 105) should be "Del14-*pep*- Δ *pepJ* (iv)", and the next entry (line 106) should be specified with "(v)".

Response:

Thank you for highlighting these issues.

Regarding Del14-*pep*- Δ *pepJ* (iv), the UPLC-HRMS analysis provided in Fig. 2a was shown as a base peak chromatogram (BPC). While the deletion of *pepJ* significantly reduced the production of pepstatins 1-4, irrelevant peaks in the crude extract eluting at the same retention times became visible, which might have caused some ambiguity. To clarify this, we have now included a direct comparison of the extracted ion chromatograms (EICs) of pepstatins from different strains in Supplementary Fig. 7a. This comparison clearly demonstrates the significant reduction in pepstatin production after *pepJ* deletion. Additionally, we acknowledge the labeling error in the figure legend. The entry "Del14-*pep*- Δ *pepJ* (v)" should indeed be corrected to "Del14-*pep*- Δ *pepJ* (iv)," and the next entry should be specified as "(v)."

The following modifications have been made to the text:

Page 4, Line 107-109:

"The production of 1-4 significantly decreased by *pepJ* deletion in Del14-*pep*- Δ *pepJ* (iv); The production of 1-4 increased in Del14-*pep*-*pepJ*-act by promoter exchange of *pepJ* (v)."

Page 5, Line 129-133:

"Deletion of the LuxR-like transcriptional regulator gene *pepJ* significantly reduced pepstatin production, suggesting that *pepJ* functions as a positive regulator. Conversely, overexpression of *pepJ* under the *kasOp* promoter led to a more than 25-fold increase in pepstatin yield (Fig. 2a and Supplementary Figs. 5–7), enabling the identification of biosynthetic intermediates in subsequent studies."

4. The NMR spectra of compound 4 looked a bit dirty. Accordingly, Figure S34-S37 (1H-NMR, 13C-NMR, HSQC, and COSY spectra of compound 4) should be supplied with the full spectra so that readers can understand the residual amount of substrates with isolated keto groups that should resonate around 210 ppm.

Response: We thank the reviewer for the insightful comment. We acknowledge the presence of some impurity signals in the NMR spectra of compound 4. However, these impurities do not compromise the structural elucidation of compound 4, as the key correlations supporting its structure remain unambiguous. Moreover, comparison with the NMR spectra of structurally highly similar compounds 1-4 further validates the deduced structure. For instance, the only structural difference between compounds 3 and 4 is the length of the fatty acid chain, differing by one CH₂ group. This difference is clearly evident in the NMR spectra: the two methyl groups (δ_{H} 1.10, δ_{C} 19.8 and δ_{H} 1.14, δ_{C} 20.1) in 3 clearly indicated an isobutyryl moiety, while the two methyl groups (δ_{H} 0.87, δ_{C} 12.4 and δ_{H} 1.11, δ_{C} 18.2) in 4 confirm the presence of a 2-methylbutanoyl unit. Additionally, MS data further supports the proposed structure of compound 4.

Notably, the observed impurity does not originate from substrates, but rather from the fermentation culture of the wild-type (WT) strain *Streptomyces catenulae* DSM40258. Despite multiple rounds of purification, some co-eluting impurities could not be completely removed. Nevertheless, as noted earlier, these impurities do not affect the structural integrity or the accuracy of the elucidated structure of compound 4. Importantly, the presence of this minor impurity does not affect the core findings of the study, which is the identification of pepstatin BGC and the characterization of the F₄₂₀H₂-dependent oxidoreductase.

Regarding the reviewer's query on keto groups resonating around 210 ppm, we confirmed that compound 4 does not contain such keto groups, as it was directly isolated from the wild-type strain. The keto groups at 210 ppm are observed in intermediates, such as compounds 9-12, produced during heterologous expression following the deletion of the oxidoreductase gene *pepI*. Additionally,

compound **13**, which also contains a keto group, was obtained through *in vitro* enzymatic reactions. For these keto-containing compounds, the relevant ^{13}C -NMR and HMBC spectra are provided in the Supplementary Information (SI).

Finally, in response to your suggestion, we have now included the full ^1H -NMR, ^{13}C -NMR, HSQC, and COSY spectra of compound **4** (Supplementary Figs. 34–37) in the SI. These additions provide a more comprehensive view of the data, allowing readers to better evaluate the residual impurities and the absence of keto groups.

5. Upon its first-time appearance, F_{420} should be defined as done elsewhere (e.g., Journal of Bacteriology 1975, 121, 154–191) so that junior readers such as graduate students can follow the story easily.

Response:

We thank the reviewer for this insightful suggestion, which will enhance the manuscript's accessibility for a broader audience. To provide context for readers unfamiliar with F_{420} , we have now introduced and defined it upon its first mention. Additionally, we have cited the recommended reference (Journal of Bacteriology, 1975, 121, 154–191), another relevant study describing the first isolation of F_{420} (Cheeseman et al., 1972, 10.1128/jb.112.1.527-531) and a review highlighted the wide spread of F_{420} and F_{420} -dependent enzymes in bacteria and archaea (Grinter & Greening, 2021, 10.1093/femsre/fuab0212021). The corresponding section in the manuscript has been revised as follows: Page 3, Line 75-78:

"In this work, we uncovered a unique post-assembly-line pathway for Sta biosynthesis that relies on a discrete oxidoreductase utilizing F_{420}H_2 , a deazaflavin-based redox cofactor widely found in bacteria and archaea. This pathway diverges from the conventional hypothesis, which attributes this function to a modular KR domain utilizing NADPH (Fig. 1c)."

We believe these additions will help readers, especially those new to the field, better follow the manuscript and appreciate the biological significance of F_{420} .

6. Many expressions/mistakes should be polished/corrected. For example, the first two sentences are not unclear enough for readers to perceive the key part of the issue the work wishes to address. To list a few more, "forms hydrogen bond interactions with" should be changed into "forms hydrogen bondings with" (line 47); "the biosynthetic pathway responsible for pepstatin production" into "the pepstatin biosynthetic pathway" (line 52); "Identification of pepstatins and the related pep biosynthetic gene cluster" into "Identification of pepstatins and the pep cluster (pep in italic)" (line 102); and in line 183, "(Fig. 11)" should be "(Supplementary Figs.11)". The authors need to re-check each sentence while improving the manuscript.

Response:

We sincerely thank you for these thoughtful suggestions, which will significantly enhance the clarity, precision, and readability of the manuscript. Following your guidance, we have carefully reviewed and revised the manuscript to address the highlighted issues and ensure the language is polished throughout. Below are the corrections made:

Page 2: the first two sentences introduce the bioactivity of aspartic proteases and highlight their significance as therapeutic targets. This information is both relevant and important to the topic of this paper—the biosynthesis of statine, the key pharmacophore responsible for pepstatin's bioactivity. Therefore, we want to retain this introduction; however, to improve clarity, we will merge the first and second paragraphs to create a smoother transition between the discussion of bioactivity and the subsequent introduction of the mode of action.

Page 2, Line 42:

We have revised "hydrogen bond interactions" to "hydrogen bonding".

Page 2, Line 47:

We changed "However, the biosynthetic pathway responsible for pepstatin production has remained unclear." to "However, the pepstatin biosynthetic pathway has remained unclear."

Page 4, Line 104:

The text has been revised for accuracy, while also reflecting the preference for concise language:
"Fig. 2 | Identification of pepstatins and the *pep* biosynthetic gene cluster."

Note: We opted to retain the term "biosynthetic gene cluster" to ensure clarity and because figure legends are designed to be standalone and comprehensible without referring to the main text.

Page 7, Line 194:

The citation error has been corrected: "(Supplementary Fig. 11b)" instead of "(Fig. 11)."

Additionally, we have rechecked the entire manuscript for other language and formatting inconsistencies to ensure the whole text is clear and professional.

Page 3, Line 72:

We have changed "hydrogen bond interactions" to "hydrogen bonds".

Page 10, Line 267; Page 11, Line 289; Page 32, Line 930:

We have changed "hydrogen bond interactions" to "hydrogen bonds".

Page 12, Line 324:

We have changed "hydrogen-bond interaction" to "hydrogen bonding".

Page 12, Line 333:

We have revised "H-bond interactions" to "hydrogen bonding".

Page 3, Line 84:

We have revised "demonstrates" to "demonstrated".

Page 6, Line 184:

We have revised "suggests" to "suggested".

Page 13, Line 355:

We have revised " the assays of kvPepI " to " kvPepI assays ".

We believe these revisions address the reviewer's concerns and greatly improve the overall quality of the manuscript.

Reviewer #2 (Remarks to the Author):

Mo et al describe the discovery of an unexpected enzyme that is essential for pepstatin biosynthesis. It is an oxidoreductase that employs F420 for the reduction of two keto groups on the statin scaffold to generate the final product. This activity is appropriately described as "tailoring". The mechanistic properties of the enzyme are analyzed in-depth and follow the expected hydride transfer scheme. The interesting feature is that the enzyme performs two reductions following a defined order. The manuscript also discusses the overall biosynthetic pathway, predicting the enzymes involved and the reactions catalyzed. However, no experimental proof is provided in support of this hypothesis.

Uncovering a new step in pepstatin biosynthesis is welcome stepforward that adds to our knowledge and understanding about the biochemistry underlying such an important class of natural products. This will also clarify past literature. The work also expands our knowledge about F420 enzymes though the reaction catalyzed by pepI is not unusual for this class of oxidoreductases. The manuscript is particularly strong with regards to the methods. It is a remarkable combination of microbial genetics, analytical chemistry, enzymology, and structural biology. The conclusions of the manuscript concerning the F420-dependent pepI therefore rely on and are corroborated by a solid experimental ground. The experimental evidence supporting the overall pathway for pepstatin biosynthesis is instead much more limited and essentially based on structural similarities. For instance, the activity of the A domain of pepD is based on "the similarity of threonine to valine" without any further element.

Response: We sincerely appreciate your positive evaluation, thoughtful comments, and constructive feedback. Indeed, the unconventional nature of the *pep* BGC presents a fascinating avenue for further investigation. However, the primary focus of this study is the identification of the pepstatin BGC and the characterization of PepI, an F₄₂₀H₂-dependent oxidoreductase, as a key enzyme responsible for iterative reductions that generate the two statine residues in pepstatin.

We acknowledge that our biosynthetic proposal is primarily based on heterologous expression results, structural considerations, and *in silico* predictions. These approaches provide a well-supported hypothesis for the pathway, yet we recognize that direct experimental evidence for each enzymatic step remains to be established. In particular, aspects such as the *trans*-acting and iterative mechanisms involving NRPS and PKS modules require further investigation, which is beyond the scope of this study. We have taken care to clearly state these limitations in the manuscript to ensure that our claims are appropriately framed.

Validating the proposed pathway, especially the noncanonical and iterative nature of the assembly line, would require extensive experimental work, including systematic *in vitro* reconstitution of all modular NRPS and PKS enzymes. Furthermore, investigating the activity of specific domains or combinations of domains within these megaenzymes would be necessary to fully understand their roles. Such studies are beyond the scope of the current work but represent important future directions that we plan to explore in subsequent research.

The primary conclusions of this study—namely, the identification of the pepstatin BGC and the characterization of PepI as a novel, F₄₂₀H₂-dependent oxidoreductase with iterative activity—are supported by a solid experimental foundation. We are grateful for the reviewer's recognition of the strengths of our methodologies and the robust data supporting the characterization of PepI.

We agree with the reviewer that a more detailed experimental characterization of the NRPS-PKS modules, including the substrate specificity of A domains and the mechanisms underlying their iterative activity, is crucial for a full elucidation of the pathway. However, achieving this level of detailed understanding would require significant additional efforts, including the *in vitro* reconstitution of the entire pathway, and is thus beyond the scope of this study.

To ensure clarity and prevent potential misunderstandings, we have carefully revised the relevant sections of the manuscript (line 135, line 138, line 144-147, line 149-155, and line 163-167) to emphasize the limitations of the current study regarding the pepstatin biosynthetic pathway. We hope that this work will serve as a foundation for future research aimed at unraveling the precise roles of NRPS-PKS modules and the overall mechanism of pepstatin biosynthesis.

Page 5, Line 135: we revised "a unique challenge" to "an interesting puzzle".

Page 5, Line 138: we revised "contradicts" to "deviates from".

Page 5. Line 144-147: we removed the sentence "Given the similarity of threonine to valine, it is plausible that ApepD selects alanine while PepG activates valine (Fig. 3)." and replaced it with "PepG is likely promiscuous in substrate activation, capable of activating threonine and similar amino acids, including valine. However, the C domains in PepH might serve as gatekeepers, processing only proteins loaded with valine." We also cite a comprehensive review paper (Bloudoff and Schmeing, 2017, 10.1016/j.bbapap.2017.05.010) which summarized the important gatekeeping function of NRPS C domains to support our proposal.

Page 5 to page 6, Line 149-152: we revised "Subsequently, PepG activates Val and loads it onto the stand-alone PCP protein PepA, with the C-terminal C domain of PepH extending the nascent chain by attacking the Val-S-PepA upon the thioester of the N-acyl-Val-S-PepH intermediate (Fig. 3)" to "A plausible subsequent step involves PepG reactivating valine and loading it onto the stand-alone PCP protein PepA. The C-terminal C domain of PepH may then facilitate chain extension by attacking the Val-S-PepA upon the thioester of the N-acyl-Val-S-PepH intermediate (Fig. 3)."

Page 6, Line 154-155: we added "may collaborate to".

Page 6. Line 163-167: we added "Notably, in pepstatin biosynthesis, this extension likely involves inter-protein interactions (Fig. 3), representing an *in-trans* mechanism distinct from intramodule interactions observed in other systems. Overall, while the proposed pathway outlines a plausible biosynthetic logic, additional experimental data are needed to validate these hypotheses and clarify the unique enzymatic features of the pepstatin biosynthetic machinery."

I have two main suggestions. The statements about the overall pathway should be rephrased to avoid any overclaiming. For instance in the discussion ("we unravel the highly disconnected and iterative NRPS-PKS hybrid biosynthetic pathway for pepstatins...The concise pepstatin assembly line stands out due to its highly disconnected nature, featuring unusual trans-acting and iterative enzymes..") as well as in the introduction ("we shed light on a noncanonical NRPS-71 PKS BGC responsible for pepstatin biosynthesis featuring in-trans and iterative NRPS and PKS, deviating significantly from the so-called "colinearity" principle...."). The manuscript puts forward a sensible and likely correct hypothesis but there is almost no data supporting it if not for the tailoring role of pepI.

Response: Thank you for your valuable suggestions. We understand your concern and have rephrased our descriptions to better reflect the data presented in the manuscript. The revised descriptions are as follows:

Page 14, Line 372-375:

"In this study, we identified the NRPS-PKS hybrid BGC responsible for pepstatin biosynthesis and revealed its distinctive disconnected and iterative utilization pattern. Most notably, we uncovered a unique F₄₂₀H₂-dependent post-assembly modification mechanism, which plays a crucial role in constructing the essential 3-OH-4-NH₂ framework."

Page 3, Line 78-81:

"Through gene knockout experiments and activation of the candidate pepstatin BGC, we characterized a noncanonical NRPS-PKS pathway responsible for pepstatin biosynthesis. Our findings suggest the

involvement of *in-trans* and iterative NRPS and PKS mechanisms, indicating deviations from the traditional "colinearity" principle."

We also rewrite the whole section "A highly dissociated and nonlinear NRPS-PKS pathway for pepstatin biosynthesis" to tone down the claims of the biosynthesis of pepstatins (line 135, line 138, line 144-147, line 149-155, and line 163-167). All parts that have been rephrased or added have been highlighted.

Can the Authors comment about other compounds of the statin family? Are they synthesized through the same tailoring activity discovered for pepstatin? Figure 1a shows many compounds and figure 4b displays some genomic comparisons. Is this a pepstatin-specific biosynthetic enzyme or is it likely to be a general feature of statin biosynthesis?

Response: Thank you for your thoughtful question. For other compounds bearing the 3-OH-4-NH₂ framework mentioned in Figure 1a, we searched for PepI homologs and F₄₂₀-dependent oxidoreductases in the available genomes of producers of didemnin, burkholdac, hapalosin, thalassospiramide, and nosperin but did not find any hits. Therefore, the most plausible hypothesis remains that the ketoreductase (KR) domain within the PKS module is responsible for the biosynthesis of Sta/Sta-like residues in these compounds.

Based on our bioinformatic analysis presented in Figure 4b, we find that PepI and its homologs from other actinomycete strains appear to play a conserved role in generating the 3-OH-4-NH₂ framework, a key feature of pepstatin. However, we also identified PepI homologs in certain *pep*-like BGCs that lack *pepC* and *pepD* homologs (Page 8, Line 201-203), leaving the products of these BGCs unclear. These findings suggest that PepI is a pepstatin-specific biosynthetic enzyme, though its presence in other *pep*-like BGCs warrants further investigation.

We added and modified several sentences as follows:

Page 2, Line 55-56: we changed "leucine" to "an α -amino acid such as leucine".

Page 2, Line 58-65:

"In the didemnin biosynthetic gene cluster (BGC), isoleucine is activated, and the ketoreductase (KR) domain within the PKS module reduces the β -keto group to form the 3-OH-4-NH₂ framework (Fig. 1c). A similar NRPS-PKS pair with a KR domain in the PKS module has also been observed in the BGCs of burkholdac, hapalosin, thalassospiramide, and nosperin. In contrast, the β -ketone functionality is retained in andrimid biosynthesis. Notably, the corresponding NRPS-PKS pair in the andrimid biosynthetic pathway lacks a KR domain, implying that the KR domain in PKS modules is responsible for 3-OH generation in Sta/Sta-like residue biosynthesis."

Page 8, Line 198-203:

"The *pep* BGC represents the first subtype, characterized by a *cis*-AT PKS gene, while the second subtype features a *trans*-AT PKS gene that lacks the AT domain. Notably, all PKS modules in these BGCs are devoid of KR domains. We therefore hypothesize that PepI analogs serve as substitutes for KR domains to generate the 3-OH-4-NH₂ moiety. The third subtype, however, lacks homologs of PepC and PepD, raising questions about its ability to produce Sta/Sta-like residues containing products (Fig. 4b)."

Page 8, line 204-207:

"Searching *pepI* homologous genes in genomes of producers of didemnin, burkholdac, hapalosin, thalassospiramide, and nosperin did not lead to any hit, further corroborating the unique Sta formation in pepstatin pathway."

Point-by-point response:

Reviewer 1

In my personal opinion, two obvious shortcomings should be improved as specified below:

1. The NMR spectra of compound **4** looked dirty, and a set of pure sample based NMR spectra (1H-NMR, 13C NMR, and 2D-NMR spectra) is desired.

Response:

We sincerely thank the reviewer for their continued critical evaluation and helpful feedback. We acknowledge that the NMR spectra of compound **4** exhibit minor impurity signals. Despite several rounds of purification using a variety of chromatographic methods, trace co-eluting impurities could not be completely removed. However, these impurities do not overlap with or obscure the key spectral features necessary for structural elucidation. The ¹H, ¹³C, and 2D NMR data remain interpretable and are consistent with the proposed structure.

Importantly, the structure of compound **4** is further supported by mass spectrometry data, as well as by comparative analysis with structurally related compounds **1–3** and **12**, for which clean NMR spectra and definitive structural assignments are provided.

We would also like to clarify that compound **4** is presented as one additional derivative in the context of a broader series. Its only structural variation lies in the N-terminal fatty acyl moiety, which does not affect the conserved core scaffold central to the biosynthetic and enzymatic findings. Therefore, the presence of minor impurities in the NMR spectra of compound **4** does not impact the validity of our structural assignments or the scientific conclusions of the study.

Finally, we emphasize that the key contributions of this work are the identification of the pepstatin biosynthetic gene cluster and the functional characterization of the F₄₂₀H₂-dependent oxidoreductase PepI. The role of compound **4** is supportive but not critical to these core findings.

We hope the reviewer finds this clarification satisfactory, and we remain grateful for their thoughtful input, which has helped improve the clarity and robustness of our manuscript.

2. The results regarding the deletion and heterologous expression of some other genes (pepA, B, C, E, F, G, H, and I) should be mentioned in the paper.

Response:

We thank the reviewer for their suggestion regarding the deletion and heterologous expression of additional genes within the *pep* cluster (pepA, B, C, E, F, G, H, and I). We assume the reviewer is referring to individual prokaryotic expression of these genes, as we have already demonstrated the successful heterologous expression of the entire *pep* biosynthetic gene cluster (BGC) in *Streptomyces albus* del14. In this context, the isolated expression of single genes in this heterologous host would not yield meaningful insights, as functional outcomes typically rely on the full assembly-line context of the BGC.

We respectfully believe that including deletion or heterologous expression data for the additional genes is not essential for the current study, for the following reasons:

1. The successful deletion of *pepD*, *pepI*, and *pepJ*, combined with the complete heterologous expression of the *pep* BGC, is sufficient to firmly establish a functional link between the *pep* cluster and pepstatin biosynthesis.
2. The primary focus of this work is the characterization of the F₄₂₀H₂-dependent oxidoreductase PepI. Additional gene deletions would not directly contribute to this central objective. In fact, disruption of genes such as *pepA*, *B*, *C*, *E*, *G*, and *H*, which are part of the NRPS-PKS assembly line, would be expected to fully abolish pepstatin production, similar to the *pepD* deletion, and thus would not yield additional mechanistic insights relevant to the current study.

We also wish to emphasize that *pepI* has already been deleted in the heterologous expression system and successfully expressed and purified as recombinant protein in *E. coli* in this study. Given this, and the focus on PepI's role, we believe the current experimental results are sufficient to support our conclusions. We appreciate the reviewer's constructive input and will consider these broader investigations in future work.